# Targeting AURKA-CDC25C axis to induce synthetic lethality in ARID1A-deficient colorectal cancer cells

Changjie Wu[1], Junfang Lyu[1], Eun Ju Yang[1], Yifan Liu[1], Baoyuan Zhang[1] & Joong Sup Shim [1]

ARID1A, a component of the SWI/SNF chromatin remodeling complex, is a tumor suppressor with a high frequency of inactivating mutations in many cancers. Therefore, ARID1A deficiency has been exploited therapeutically for treating cancer. Here we show that ARID1A has a synthetic lethal interaction with aurora kinase A (AURKA) in colorectal cancer (CRC) cells. Pharmacological and genetic perturbations of AURKA selectively inhibit the growth of ARID1A-deficient CRC cells. Mechanistically, ARID1A occupies the *AURKA* gene promoter and negatively regulates its transcription. Cells lacking ARID1A show enhanced *AURKA* transcription, which leads to the persistent activation of CDC25C, a key protein for G2/M transition and mitotic entry. Inhibiting AURKA activity in ARID1A-deficient cells significantly increases G2/M arrest and induces cellular multinucleation and apoptosis. This study shows a novel synthetic lethality interaction between ARID1A and AURKA and indicates that pharmacologically inhibiting the AURKA–CDC25C axis represents a novel strategy for treating CRC with *ARID1A* loss-of-function mutations.

[1] Faculty of Health Sciences, University of Macau, Avenida da Universidade, 999078 Taipa, Macau SAR, China. Correspondence and requests for materials should be addressed to J.S.S. (email: jsshim@umac.mo)

The SWI/SNF chromatin remodeling complex remodels nucleosomes and modulates transcription in an ATP-dependent manner[1]. This complex exists as two major forms, BRG1-associated factor (BAF) and polybromo BAF[2]. Each complex contains 8–15 subunits, and many subunits have multiple isoforms. Mutations in these subunits lead to the aberrant control of lineage-specific differentiation and gene expression/repression, thereby contributing to tumorigenesis; these mutations have been observed in a number of cancer types[1]. AT-rich interactive domain 1A (ARID1A), a component of the BAF complex, has been identified by next-generation sequencing as one of the most frequently mutated genes in a variety of cancers, including ovarian clear cell carcinoma (OCCC)[3], gastric cancer[4], hepatocellular carcinoma[5], esophageal adenocarcinoma[6], breast cancer[7], pancreatic cancer[8] and colorectal cancer (CRC)[9]. In addition, loss of ARID1A expression has also been observed in different cancer types, such as uterine endometrioid carcinoma[10] and renal cancer[11]. Genome-wide sequencing analyses of tumor samples revealed that 46–57% of OCCC cases harbored loss-of-function mutations in the ARID1A gene, implying the significant contribution of aberrant ARID1A functions to OCCC pathogenesis[3,12]. In CRC patients, a mutation frequency of approximately 10% was observed for the ARID1A gene[13]. However, clinico-pathological analyses of ARID1A protein levels in CRC tumor samples showed that 25.8% of CRC primary tumors did not express ARID1A, and 51.2% had low expression levels of ARID1A (77% of all the CRC samples had no or low ARID1A expression)[14]. The loss of ARID1A expression became even more significant as the tumor–node–metastasis (TNM) stage advanced. ARID1A loss was observed for 7.4% of TNM stage I samples, 24.1% of TNM stage II samples, 22.2% of TNM stage III samples, and 46.3% of TNM stage IV samples[14]. These data suggest that ARID1A loss in CRC is strongly associated with tumor progression and metastasis.

Since the discovery of the high frequency of mutations and loss of expression of ARID1A in cancer, ARID1A deficiency has been exploited therapeutically for treating cancer according to an approach called synthetic lethality. Synthetic lethality is a genetic interaction between two or more genes where a single gene deficiency does not affect cell viability, but the combination of both gene deficiencies causes lethality. This concept has been widely exploited in cancer therapy because many types of cancer have loss-of-function mutations in tumor-suppressor genes that are not readily targetable. The pharmacological or genetic disruption of a synthetic lethality target of a tumor suppressor will cause selective lethality in the cancer cells that harbor the tumor-suppressor mutations[15]. Recent studies have shown that ARID1A has a synthetic lethality interaction with genes involved in some epigenetic machinery, including EZH2[16], poly ADP-ribose polymerase 1 (PARP1)[17], ATR[18], and histone deacetylase 6 (HDAC6)[19]. Inhibiting the synthetic lethality targets resulted in selective vulnerabilities in ARID1A mutant OCCC, CRC, and breast cancer cells[16–19]. These studies suggested that ARID1A, as an epigenetic machinery component, may have various genetic and functional interdependencies with other epigenetic components to affect cell survival. Based on this notion, we initiated a systematic screening for druggable targets among human epigenetic machinery using an ARID1A isogenic CRC pair and epigenetics drug library. Among the epigenetics drugs screened, aurora kinase A (AURKA) inhibitors composed the majority of the synthetic lethality hits.

AURKA, also known as serine/threonine protein kinase 6, is a member of the mitotic serine/threonine kinase family, which has multiple functions in mitosis and non-mitotic biological processes[20–22]. During mitosis, AURKA phosphorylates several substrates, including polo-like kinase 1 (PLK1), to promote entry into mitosis at the G2/M phase by activating the nuclear localization of cell division cycle 25C (CDC25C)[23,24]. AURKA overexpression has been implicated in genetic instability and tumorigenesis[25], which are observed in many cancers, including leukemia[26], ovarian[27], lung[28], pancreas[29], liver[30], and CRC[31]. High AURKA expression has been associated with poor overall survival in patients with metastatic CRC[32] and non-small cell lung cancer[33], suggesting that it is an important therapeutic target for developing anticancer drugs.

In this study, we show that AURKA inhibition causes selective vulnerability in CRC cells lacking ARID1A. We further explore a mechanism whereby the ARID1A and AURKA pathways converge on CDC25C to induce G2/M arrest and apoptosis in CRC cells.

## Results

**ARID1A synthetic lethality drug screening in CRC cells.** To screen and identify ARID1A synthetic lethality targets, we first generated ARID1A isogenic CRC pairs using a clustered regularly interspaced short palindromic repeats (CRISPR)/Cas9-mediated gene knockout (KO). The use of ARID1A isogenic cell pairs will ensure that the synthetic lethality identified is dependent on the ARID1A status rather than other cellular factors. HCT116 CRC cells carrying wild-type ARID1A (ARID1A$^{+/+}$) were transfected with a Cas-9 plasmid, single guide RNAs (sgRNAs) targeting the ARID1A gene and HDR (homology-directed repair) donor plasmids containing puromycin-resistant and red fluorescence protein (RFP) genes (Supplementary Fig. 1a). The KO clones were selected for RFP fluorescence and puromycin resistance (Supplementary Fig. 1b, c). ARID1A KO was confirmed with genomic PCR sequencing and immunoblot analysis (Fig. 1a–c and Supplementary Fig. 1d). The sequencing analysis showed that some KO clones had a homozygous HDR insertion in both ARID1A alleles, and others had heterozygous mutations (an HDR insertion in one allele and a non-homologous end joining (NHEJ)-mediated indel mutation in the other) (Fig. 1b). Among the three confirmed ARID1A KO clones (ARID1A$^{-/-}$ #1–3) (Fig. 1c), ARID1A$^{-/-}$ #1 was used for the synthetic lethality screening and the other clones were used to validate the screening hits. In vitro growth rates for the ARID1A$^{+/+}$ and ARID1A$^{-/-}$ HCT116 CRC cells were similar in short-term culture (Supplementary Fig. 2a), which was in agreement with previous reports[16,19,34]. To screen for druggable human epigenetic protein targets, we used a human epigenetics compound library containing 128 small-molecule inhibitors targeting all known druggable epigenetic proteins. The screening was done with an 8-dose interplate titration format in 384-well plates to determine the estimated IC$_{50}$ values of each compound for the ARID1A isogenic pair (ARID1A$^{+/+}$ and ARID1A$^{-/-}$ #1 cells) (Fig. 1d). From two rounds of screening, we identified 6 candidate drugs that showed a selectivity index (SI) >2 for the ARID1A$^{-/-}$ #1 cells; the candidates included 3 AURKA inhibitors (Aurora A inhibitor I, MK-5108 and JNJ-7706621), a histone demethyltransferase inhibitor (GSK J4), a PARP inhibitor (PJ34), and a histone methyltransferase inhibitor (BIX 01294) (Fig. 1e; Supplementary Fig. 2c and d). Since the majority of the identified candidates were AURKA inhibitors (3 out of 6), we selected Aurora A inhibitor I (AURKAi) as the primary synthetic lethality compound for follow-up studies. AURKAi treatment showed decent selectivity toward all three ARID1A$^{-/-}$ clones compared to wild-type HCT116 cells (Fig. 1f). We also compared the synthetic lethality of AURKAi with that of other known ARID1A synthetic lethality targets, including HDAC6[19], ATR[18], PARP[17], and EZH2[16]. The synthetic lethality of AURKAi was largely comparable with that of HDAC6, ATR, and PARP inhibitors (Fig. 1g–j). The EZH2 inhibitor EPZ-6438

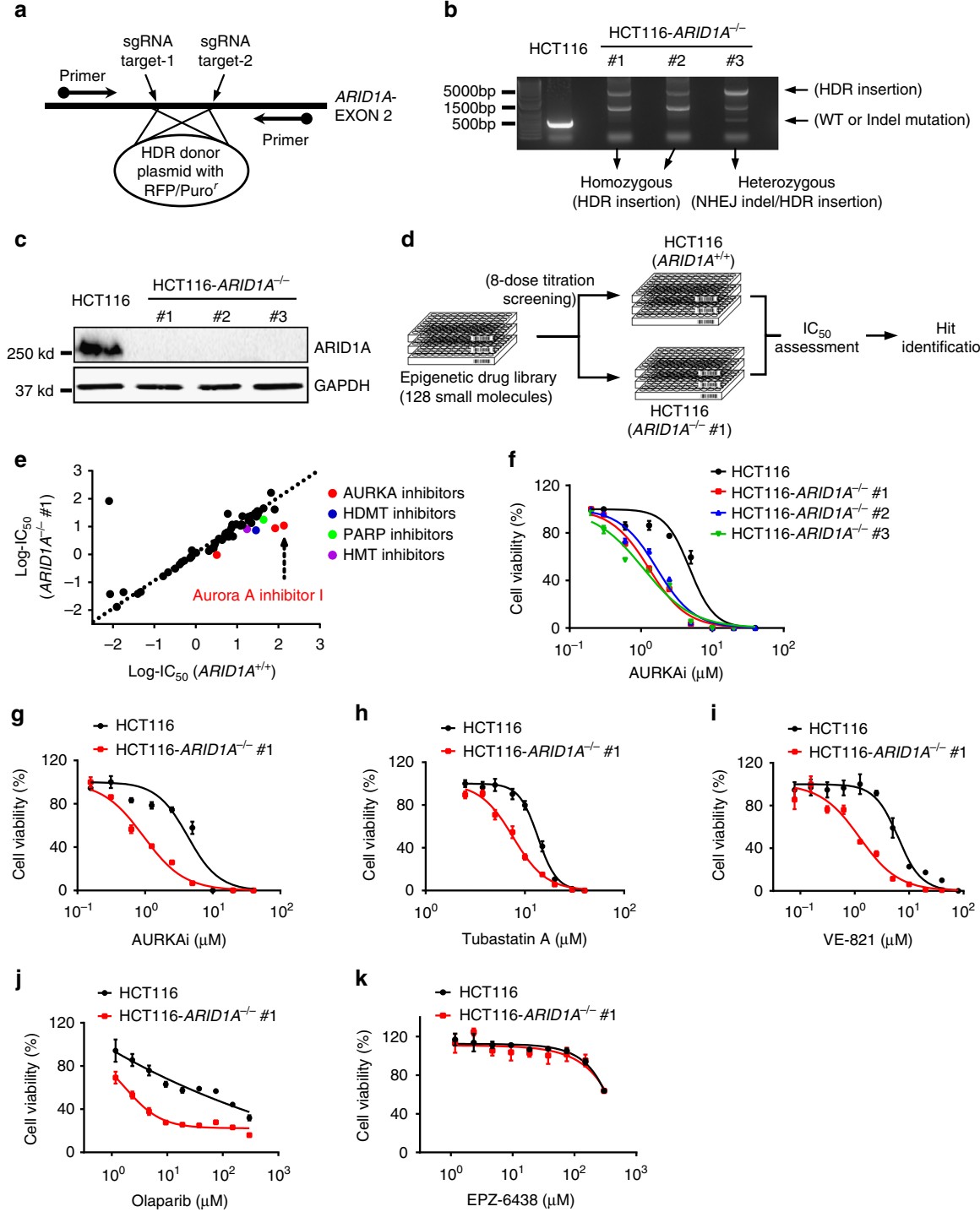

did not have synthetic lethality effects on the HCT116 CRC isogenic pair (Fig. 1k), even with longer (96 h) treatment (Supplementary Fig. 2e). This effect was possibly because ARID1A deficiency in CRC cells did not reduce the expression of phosphoinositide-3 kinase-interacting protein 1 (PIK3IPI), which is a key target gene for the synthetic lethality of EZH2 inhibitors in ovarian cancer cells (Supplementary Fig. 2f); alternatively, the effect could be due to the activating mutation in *PIK3CA* in HCT116 cells[35].

**ARID1A has a synthetic lethal interaction with AURKA**. To further examine the synthetic lethality of AURKAi, nuclear

morphology was analyzed in *ARID1A* isogenic cells treated with AURKAi. AURKAi treatment had negligible effects on *ARID1A* wild-type cells but markedly increased nuclear condensation and fragmentation, which are typical signs of apoptotic cell death, in all three *ARID1A*$^{-/-}$ clones (Fig. 2a, b). To determine whether AURKA is the protein responsible for the synthetic lethality, we transfected *AURKA* small interference RNA (siRNA) into the *ARID1A* isogenic cell pair, and cell viability was assessed. Similar to what was seen with AURKAi treatment, silencing *AURKA* with siRNA selectively inhibited viability in *ARID1A*$^{-/-}$ cells (Fig. 2c–e). To further test whether the synthetic lethality induced by AURKA inhibition was dependent on ARID1A status, we

ectopically expressed wild-type *ARID1A* in *ARID1A*$^{-/-}$ cells and examined the synthetic lethality. Re-expression of wild-type *ARID1A* in *ARID1A*$^{-/-}$ cells significantly reversed the inhibition of cell viability by AURKAi (Fig. 2f, g). These results suggest that AURKA has a synthetic lethal interaction with ARID1A in CRC cells. We next examined the synthetic lethality of AURKAi in a tumor xenograft mouse model. Mice bearing *ARID1A* isogenic tumors on either flank were given AURKAi via intraperitoneal (i. p.) injection, and the tumor volumes were measured periodically (Fig. 2h). AURKAi treatment did not affect the growth of *ARID1A* wild-type tumor xenografts (Fig. 2i, k), but it significantly inhibited the growth of *ARID1A*$^{-/-}$ tumor xenografts at the same dosage (Fig. 2j, l). Immunostaining with the cell proliferation marker Ki67 also showed the selective antitumor activity of AURKAi against *ARID1A*$^{-/-}$ tumors (Supplementary Fig. 3a). The daily administration of 30 and 60 mg kg$^{-1}$ AURKAi did not appear to cause toxicity in mice as assessed by body weight changes (Supplementary Fig. 3b).

Next, we tested the synthetic lethality of AURKA inhibition in CRC cells derived from *ARID1A* mutant tumors. The RKO CRC cell line has a frameshift deletion mutation in *ARID1A*[36,37]. Therefore, we used RKO cells to generate another *ARID1A* isogenic pair by overexpressing wild-type *ARID1A* via lentivirus in RKO cells (Fig. 3a). Similar to the HCT116 *ARID1A* isogenic pair, the growth rates of RKO parental and ARID1A-expressing clones were largely similar (Supplementary Fig. 2b). Using the RKO-*ARID1A* isogenic cell pair, we further validated that either AURKAi treatment (Fig. 3b) or *AURKA* silencing by siRNA transfection (Fig. 3c, d) selectively inhibited viability in ARID1A-deficient CRC cells. Synthetic lethality in the RKO-*ARID1A* isogenic cell pair was further tested in the tumor xenograft mouse model (Fig. 3e). AURKAi treatment significantly inhibited the growth of *ARID1A*$^{-/-}$ RKO tumors (Fig. 3f, h), while it had no significant effect on the growth of ARID1A-expressing RKO tumors (Fig. 3g, i). Taken together, these results demonstrated that AURKA is a synthetic lethality target in ARID1A-deficient CRC cells and that AURKA inhibition-induced synthetic lethality is dependent on ARID1A status.

Because many of the *ARID1A* mutations in patients are heterozygous (*ARID1A*$^{+/-}$)[3,38], it is important to test whether the synthetic lethality of AURKAi is also effective in cells with heterozygous mutations. Since we failed to isolate *ARID1A*$^{+/-}$ clones from the HCT116 *ARID1A* KO study, we knocked out *ARID1A* in SW480 cells, an *ARID1A*$^{+/+}$ CRC cell line. Through CRISPR/Cas-9 gene editing, we successfully generated both homozygous (*ARID1A*$^{-/-}$) and heterozygous (*ARID1A*$^{+/-}$) KO clones from SW480 cells (Supplementary Fig. 4a, b). The SW480 *ARID1A*$^{+/-}$ B9 clone had barely detectable ARID1A protein expression, which was in agreement with previous reports that > 70% of heterozygous *ARID1A* mutations lack protein expression[3,38]. This clone and the two homozygous *ARID1A*$^{-/-}$ clones were more sensitive to AURKAi treatment than the parental SW480 cells expressing wild-type *ARID1A* (Supplementary Fig. 4c). These data suggest that the synthetic lethality of ARID1A and AURKA is largely common among CRC cells and that the synthetic lethality is applicable to heterozygous *ARID1A* mutations with a loss of protein expression. Since *ARID1A* mutations are highly common in OCCC, we next tested synthetic lethality in different subtypes of ovarian carcinoma cell lines. *ARID1A* mutant SKOV3 cells were originally described as high-grade serous carcinoma, but it was recently re-described as OCCC according to histological and immunological characterizations of in vivo tumors[39,40]. ES-2 cells express wild-type *ARID1A* and were originally described as OCCC[41,42]. HO8910 cells characterized as ovarian serous carcinoma with wild-type *ARID1A*. Our results showed that ARID1A-deficient SKOV3 cells were significantly more sensitive to AURKAi treatment than *ARID1A* wild-type HO8910 or ES-2 cells (Supplementary Fig. 5a–c). These data suggested that ARID1A–AURKA synthetic lethality exists in ovarian cancer cells and is dependent on ARID1A status rather than on tumor subtype.

**AURKAi induces G2/M arrest and apoptosis in *ARID1A*$^{-/-}$ cells.** We next analyzed the cellular phenotype of CRC upon AURKA inhibition. During mitosis, AURKA functions mainly in centrosome separation and maturation to help assemble bipolar spindles[43]. Knocking down *AURKA* by siRNA caused abnormal chromosome arrangement and segregation in CRC cells (Fig. 4a). This effect was more severe in *ARID1A*$^{-/-}$ cells where spindle pole fragmentation was observed. Interestingly, the number of multinucleated cells was higher in the *ARID1A*$^{-/-}$ group than in the wild-type group (Fig. 4b, c). Treatment with *AURKA* siRNA further increased the number of multinucleated cells, with the highest proportion in the *ARID1A*$^{-/-}$ group (Fig. 4b, c). Similarly, treating CRC cells with AURKAi caused G2/M cell cycle arrest and increased the percentage of tetraploid cells, especially in the *ARID1A*$^{-/-}$ group (Fig. 4d, e). This effect was accompanied by a significant increase in the sub-G1 population, an indicator of apoptosis, in *ARID1A*$^{-/-}$ cells (Fig. 4d). Indeed, the genetic and pharmacological inhibition of AURKA selectively induced apoptosis in *ARID1A*$^{-/-}$ HCT116 cells (Fig. 4f–i). The selective induction of apoptosis in ARID1A-deficient cancer was further confirmed in the *ARID1A* isogenic RKO xenograft mouse model (Supplementary Fig. 3c). These results suggest that ARID1A deficiency causes abnormal cell division, while AURKA inhibition leads to a defect in bipolar spindle assembly and G2/M

**Fig. 1** Generation of ARID1A-knockout (KO) HCT116 cells and screening of epigenetic drug library for synthetic lethality. **a** Illustration for sgRNA target sites on *ARID1A* exon 2 for HDR donor plasmid insertion and primers designed for Sanger sequencing. **b** PCR amplification of *ARID1A* exon 2 using the designed primers in HCT116 *ARID1A*$^{+/+}$ and three *ARID1A*$^{-/-}$ clones. *ARID1A*$^{-/-}$ clones #1 and #2 are homozygous ARID1A-KO with HDR donor plasmid insertion into *ARID1A* gene. Clone #3 is a heterozygous *ARID1A*$^{-/-}$ containing a HDR insertion mutation and an NHEJ Indel mutation. **c** Immunoblot analysis showing loss of ARID1A expression in the three *ARID1A*$^{-/-}$ clones. **d** Schematic illustration of the synthetic lethality epigenetic drug screening. HCT116 *ARID1A*$^{+/+}$ and *ARID1A*$^{-/-}$ #1 cell lines were screened in parallel with 128 epigenetic drug library in an 8-dose titration format. After incubation with the drug library for 72 h, cell viability was determined by AlamarBlue assay. **e** A log10-IC$_{50}$ plot of the screening results. A log10 scale of IC$_{50}$ values of the drugs against HCT116 *ARID1A*$^{+/+}$ and *ARID1A*$^{-/-}$ cells was plotted. Drugs with selectivity index (SI) > 2 for ARID1A$^{-/-}$ cells were selected and marked as synthetic lethality candidates. **f** Dose–response curves of HCT116 *ARID1A*$^{+/+}$ and three *ARID1A*$^{-/-}$ clones treated with Aurora A inhibitor I (AURKAi) for 72 h are shown. Error bars represent s.d. (*n* = 9) from three independent experiments. Survival curve of all three KO clones versus wild-type cells *P* value < 0.0001, ANOVA. **g–k** Dose–response curves of HCT116 *ARID1A* isogenic cell pair treated with AURKAi (**g**) and known synthetic lethality compounds for ARID1A, including tubastatin A (HDAC6 inhibitor) (**h**), VE-821 (ATR inhibitor) (**i**), olaparib (PARP inhibitor) (**j**), and EPZ-6438 (EZH2 inhibitor) (**k**), are shown. HCT116 *ARID1A*$^{+/+}$ and *ARID1A*$^{-/-}$ #1 clone were incubated with indicated compounds for 72 h and the cell viability was determined by AlamarBlue assay. Error bars represent s.d. (*n* = 9) from three independent experiments. ANOVA *P* value of <0.0001 for AURKAi, tubastatin A, VE-821, and olaparib. ANOVA *P* value of 0.1629 for EPZ-6438

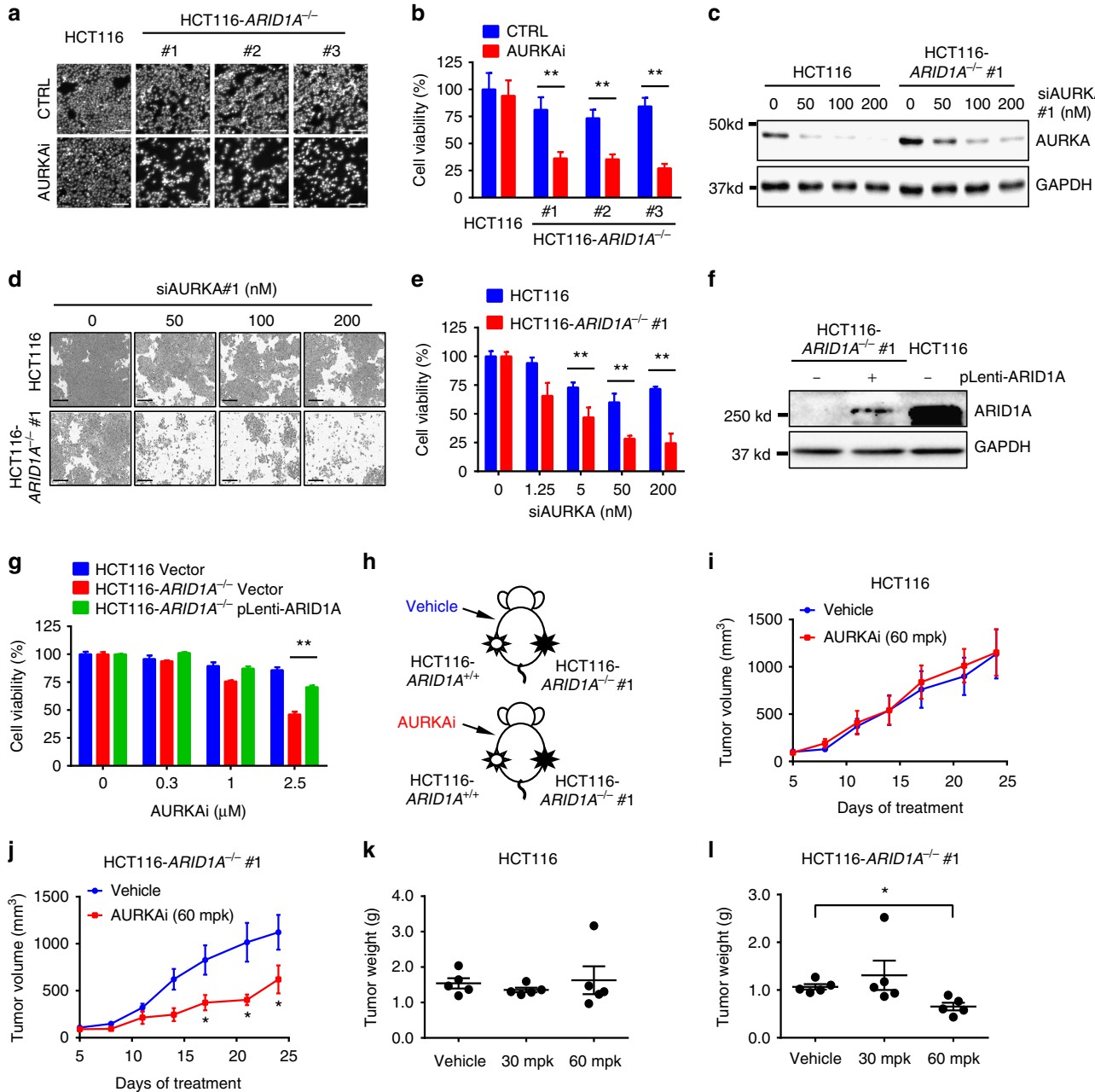

**Fig. 2** In vitro and in vivo synthetic lethality in ARID1A-KO HCT116 cells by AURKA inhibition. **a** Synthetic lethality in *ARID1A*−/− HCT116 cells by AURKAi. *ARID1A*+/+ HCT116 or three *ARID1A*−/− clones were treated with 1 μM AURKAi for 72 h and nuclei were stained with Hoechst 33342. Scale bars, 100 μm. **b** Nuclear density was measured with Image J software as a surrogate for cell viability. Error bars represent s.d. **$P < 0.01$, Student's *t* test. **c** Silencing of *AURKA* expression in *ARID1A* isogenic cell pair by siRNA (siAURKA#1). GAPDH was used as a loading control. **d** Synthetic lethality in *ARID1A*−/− HCT116 cells by *AURKA* siRNA. *ARID1A*+/+ or *ARID1A*−/− clone was transfected with various concentrations of *AURKA* siRNA for 72 h and the cell images were taken with IncuCyte ZOOM. Scale bars, 300 μm. **e** Integrated density was measured with the IncuCyte ZOOM software as a surrogate for cell viability. Error bars represent s.d. **$P < 0.01$, Student's *t* test. **f** Ectopic overexpression of ARID1A (pLenti-ARID1A) in *ARID1A*−/− HCT116 cells. **g** Overexpression of ARID1A reversed the synthetic lethality effect by AURKAi. *ARID1A*+/+ HCT116, *ARID1A*−/− HCT116, and *ARID1A*−/− HCT116 transfected with an ARID1A plasmid were treated with AURKAi for 72 h, and the cell viability was assessed with AlamarBlue assay. Error bars represent s.d. **$P < 0.01$, Student's *t* test. **h** Schematic illustration of mouse tumor xenograft experiments with HCT116 *ARID1A* isogenic cell pair. **i**, **j** Tumor growth curve in nude mice bearing *ARID1A*+/+ HCT116 (**i**) or *ARID1A*−/− HCT116 (**j**) xenografts after injection of vehicle or 60 mg kg−1 (mpk) AURKAi. Error bars represent s.d. *$P < 0.05$ between vehicle and AURKAi treatment groups ($n = 5$), Student's *t* test. **k**, **l** Wet weight measurement of the tumors isolated from mice bearing *ARID1A*+/+ HCT116 (**k**) or *ARID1A*−/− HCT116 (**i**) xenografts at 24 days after injection of vehicle, 30 or 60 mpk AURKAi. Error bars represent s.d. *$P < 0.05$ between vehicle and 60 mpk AURKAi treatment groups ($n = 5$), Student's *t* test

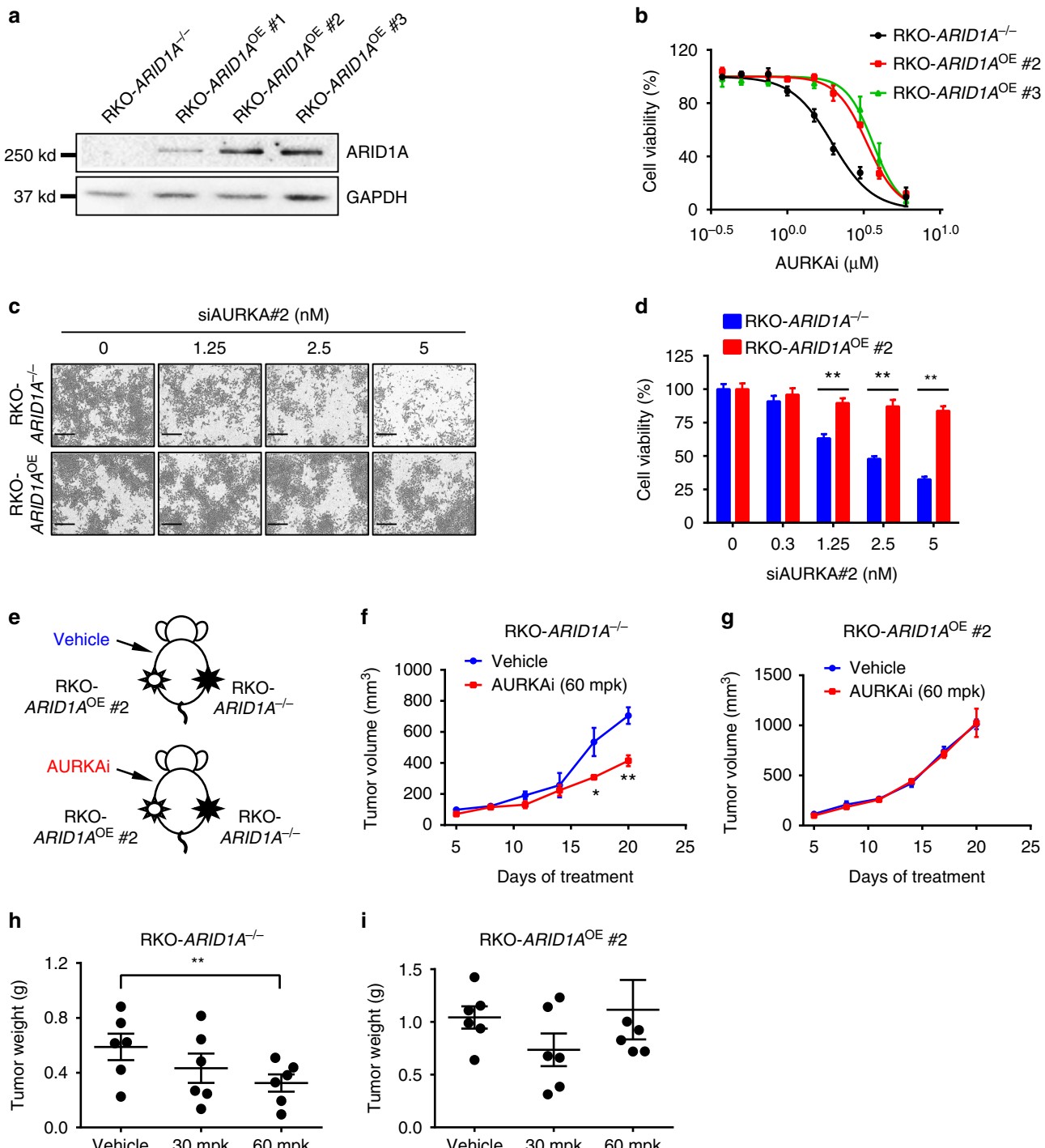

**Fig. 3** In vitro and in vivo synthetic lethality in *ARID1A*−/− RKO cells by AURKA inhibition. **a** Stable expression of ARID1A in *ARID1A*−/− RKO cells using a lentiviral transduction. Three selected ARID1A stable clones are shown. **b** Dose–response curves of *ARID1A*−/− parental RKO and ARID1A overexpressing (ARID1A^OE) RKO clones treated with AURKAi. Error bars represent s.**d**. (*n* = 6) from three independent experiments. Survival curve of *ARID1A*−/− versus ARID1A^OE cell lines, *P* value < 0.0001, ANOVA. **c** Synthetic lethality effect of *AURKA* siRNA (siAURKA#2) on RKO *ARID1A* isogenic pair. Representative cell images were taken with IncuCyte ZOOM. Scale bars, 300 μm. **d** Integrated cell density was measured with the IncuCyte ZOOM software as a surrogate for cell viability (right panels). Error bars represent s.d. **P < 0.01, Student's *t* test. **e** Schematic illustration of mouse tumor xenograft experiments with RKO *ARID1A* isogenic cell pair. **f**, **g** Tumor growth curve in nude mice bearing *ARID1A*−/− RKO (**f**) or ARID1A^OE RKO clone #2 (**g**) xenografts after injection of vehicle or 60 mg kg−1 (mpk) AURKAi. Error bars represent s.d. *P < 0.05; **P < 0.01 between vehicle and AURKAi treatment groups (*n* = 6), Student's *t* test. **h**, **i** Wet weight measurement of the tumors isolated from mice bearing *ARID1A*−/− RKO (**h**) or ARID1A^OE RKO clone #2 (**i**) xenografts at 20 days after injection of vehicle, 30 or 60 mpk AURKAi. Error bars represent s.d. **P < 0.01 between vehicle and 60 mpk AURKAi treatment groups (*n* = 6), Student's *t* test

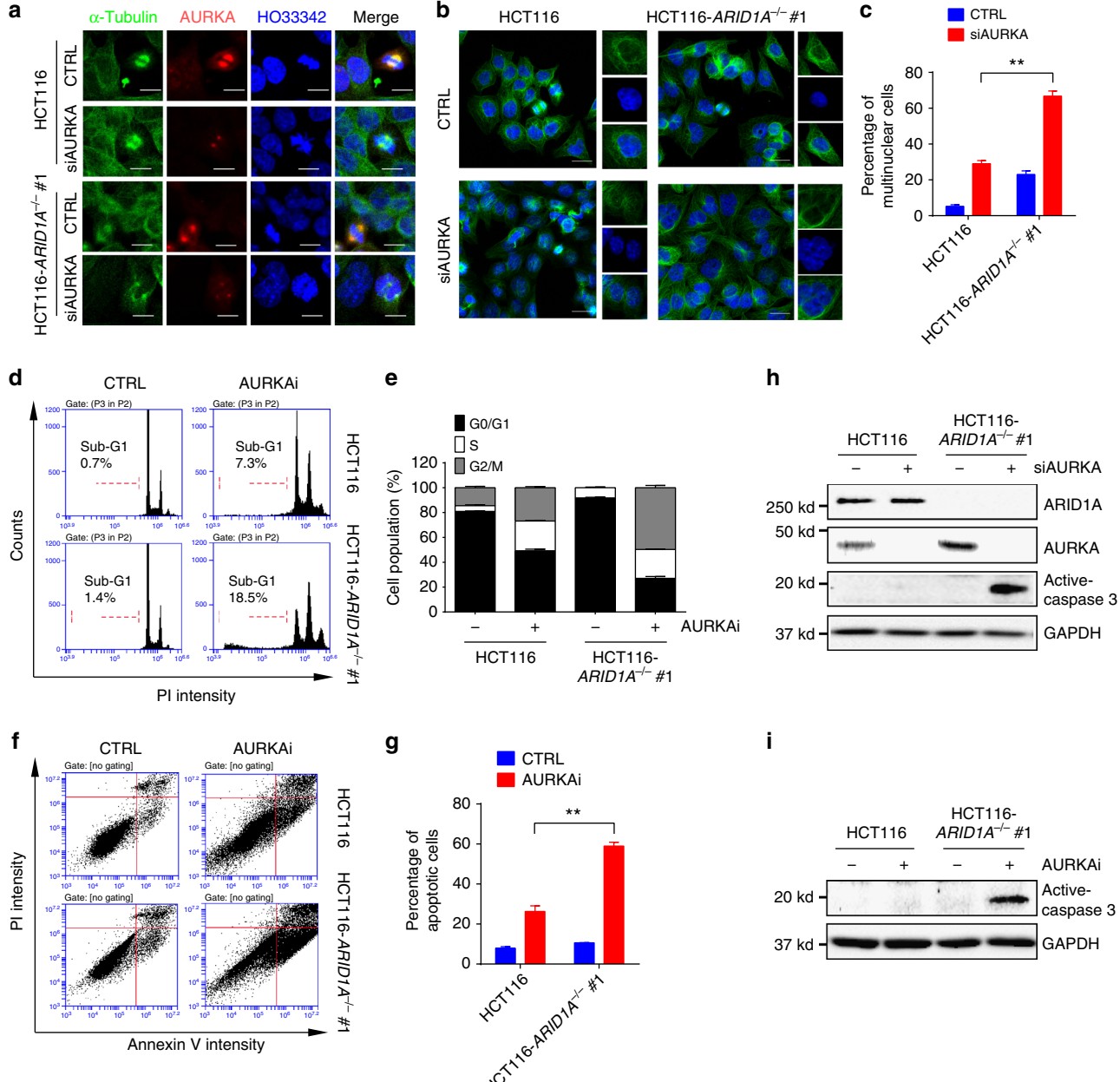

**Fig. 4** Induction of multinucleation, G2/M arrest, and apoptosis in *ARID1A*$^{-/-}$ cells by AURKA inhibition. **a** Abnormal chromosomal segregation induced by *AURKA* silencing. *ARID1A* isogenic HCT116 cells were transfected with *AURKA* siRNA and analyzed for immunofluorescence of AURKA, α-tubulin, and nuclei/chromatin (HO33342). Scale bars, 10 µm. **b** Induction of multinucleation in *ARID1A*$^{-/-}$ cells by *AURKA* silencing. *ARID1A* isogenic HCT116 cells were transfected with *AURKA* siRNA and analyzed for immunofluorescence of α-tubulin (green) and nuclei/chromatin (blue). Images in the inlets (red square) are representative cells that were magnified and shown on the right side of each figure. Scale bars, 20 µm. **c** Percentage of multinuclear cells in *ARID1A* isogenic HCT116 cells treated with *AURKA* siRNA. Error bars represent s.d. **$P < 0.01$, Student's *t* test. **d** Cell cycle analyses of *ARID1A* isogenic HCT116 cells treated with AURKAi. **e** Percentage of cell populations in each cell cycle phase. Error bars represent s.d. **f**, **g** Preferential induction of apoptosis in *ARID1A*$^{-/-}$ cells by AURKAi. Cells with or without AURKAi treatment were subjected to FITC–Annexin V apoptosis detection with a flow cytometer (**f**) and the percentage of apoptotic cells were quantitated (**g**). Error bars represent s.d. **$P < 0.01$, Student's *t* test. **h**, **i** Preferential induction of apoptosis in *ARID1A* $^{-/-}$ cells by *AURKA* silencing (**h**) or AURKAi treatment (**i**). Active (cleaved) caspase-3 was used as a marker of apoptosis induction

arrest. The combination of these two effects in *ARID1A*$^{-/-}$ cells likely caused the increase in multinucleated cells, thereby inducing cell apoptosis.

**ARID1A represses *AURKA* transcription in CRC cells**. We next investigated the mechanism of the synthetic lethality interaction between ARID1A and AURKA in CRC cells. In this study, we noted that AURKA expression was upregulated in *ARID1A*$^{-/-}$

cells (Fig. 2c). Therefore, we investigated the possibility that ARID1A regulated the gene expression of AURKA. Indeed, we observed a significant increase in AURKA protein levels in all HCT116 *ARID1A*$^{-/-}$ clones and ARID1A-deficient RKO cells compared to those in the corresponding wild-type *ARID1A*-expressing cells (Fig. 5a, b). To determine whether the AURKA upregulation in ARID1A-deficient cells was due to the dynamic gene expression control by ARID1A, we silenced *ARID1A*

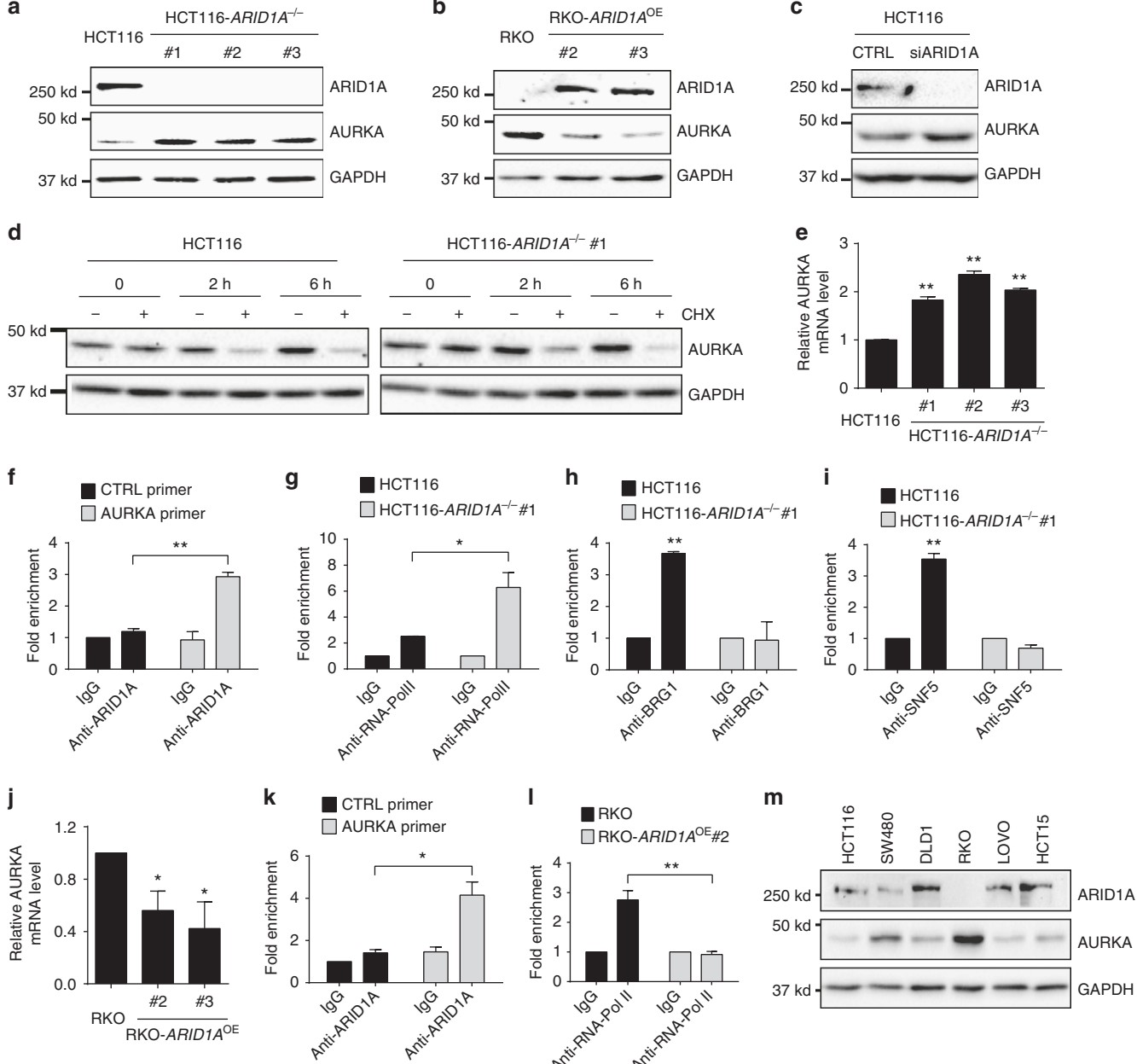

**Fig. 5** *AURKA* is a target gene for transcription repression by ARID1A. **a** Upregulation of AURKA level in *ARID1A*$^{-/-}$ HCT116 cells. **b** Downregulation of AURKA level in ARID1A overexpressing (ARID1A$^{OE}$) RKO clones. **c** Upregulation of AURKA level in HCT116 cells by *ARID1A* silencing. **d** Measurement of AURKA protein half-life in *ARID1A* isogenic HCT116 cells. Cells were treated with or without 10 μM cycloheximide (CHX) for the indicated time points and pre-existing AURKA protein stability was analyzed. **e** RT-qPCR analysis of *AURKA* mRNA level in HCT116 *ARID1A*$^{+/+}$ and *ARID1A*$^{-/-}$ clones. **P < 0.01 vs HCT116 (*ARID1A*$^{+/+}$), one-sample *t* test. **f** Chromatin immunoprecipitation (ChIP) of *AURKA* promoter in *ARID1A*$^{+/+}$ HCT116 cells using anti-ARID1A antibody. ChIP data were normalized to the control IgG ChIP with the control (CTRL) primer. **P < 0.01, Student's *t* test. **g** ChIP of *AURKA* promoter in HCT116 *ARID1A*$^{+/+}$ (black) and *ARID1A*$^{-/-}$ (gray) cells using anti-RNA-Pol II antibody. IgG in each cell line was used as a normalization control. *P < 0.05, Student's *t* test. **h** ChIP of *AURKA* promoter in HCT116 *ARID1A*$^{+/+}$ (black) and *ARID1A*$^{-/-}$ (gray) using anti-BRG1 antibody. IgG in each cell line was used as a normalization control. **P < 0.01 vs IgG, one-sample *t* test. **i** ChIP of *AURKA* promoter in HCT116 *ARID1A*$^{+/+}$ (black) and *ARID1A*$^{-/-}$ (gray) using anti-SNF5 antibody. IgG in each cell line was used as a normalization control. **P < 0.01 vs IgG, one-sample *t* test. **j** RT-qPCR analysis of *AURKA* mRNA level in RKO *ARID1A*$^{-/-}$ and ARID1A$^{OE}$ clones. *P < 0.05 vs RKO (*ARID1A*$^{-/-}$), one-sample *t* test. **k** ChIP of AURKA promoter in ARID1A$^{OE}$ RKO clone #2 using anti-ARID1A antibody. ChIP data were normalized to the control IgG ChIP with the control (CTRL) primer. *P < 0.05, Student's *t* test. **l** ChIP of *AURKA* promoter in RKO *ARID1A*$^{-/-}$ (black) and ARID1A$^{OE}$ (gray) cells using anti-RNA-Pol II antibody. IgG in each cell line was used as a normalization control. **P < 0.01, Student's *t* test. Error bars represent s.d. **m** Protein expression status of ARID1A and AURKA in six colorectal cancer cell panel

expression using siRNA and analyzed AURKA levels. Similar to the results in *ARID1A*$^{-/-}$ cells, *ARID1A* silencing also increased the AURKA protein levels (Fig. 5c). Since AURKA protein levels can be changed either transcriptionally or posttranslationally, we

first determined the apparent half-life of the AURKA protein in the *ARID1A* isogenic cell pair. Cycloheximide treatment showed that the half-life of AURKA was similar in *ARID1A*$^{+/+}$ and *ARID1A*$^{-/-}$ cells, and the estimated half-lives were 2 h (Fig. 5d);

this finding suggested that ARID1A did not affect AURKA protein stability. We next examined the transcription levels of AURKA in the *ARID1A* isogenic cell pairs. Reverse transcriptase–quantitative real time PCR (RT-qPCR) analysis showed that all three of the HCT116 *ARID1A*$^{-/-}$ clones exhibited a significant increase in *AURKA* mRNA levels (Fig. 5e), demonstrating that ARID1A negatively regulates *AURKA* transcription. This notion was further evidenced by the down-regulation of *AURKA* mRNA in ARID1A-overexpressing RKO cell clones (Fig. 5j). Since the SWI/SNF complex contributes to either gene activation or repression by remodeling the nucleosome position[1], it is possible that ARID1A may directly repress *AURKA* transcription by occupying its promoter. We therefore conducted chromatin immunoprecipitation (ChIP) assays of the *AURKA* promoter using specific antibodies against ARID1A and RNA polymerase II (RNA-Pol II) and a primer pair that targets the transcription start site (TSS) of the *AURKA* gene (Primer #1; Supplementary Fig. 6a). Indeed, the *AURKA* promoter region was significantly enriched in the ChIP assays with an anti-ARID1A antibody (Fig. 5f). In contrast, no significant enrichment of ARID1A was observed with a control primer pair at the open reading frame (ORF)-free region (Fig. 5f) or with primer pairs at >3 kb upstream and downstream of the *AURKA* TSS (Primer #2 and #3; Supplementary Fig. 6a and b). In addition, RNA-Pol II bound predominantly to the *AURKA* promoter in *ARID1A*$^{-/-}$ cells (Fig. 5g), indicating that *AURKA* is transcriptionally active in *ARID1A*$^{-/-}$ cells. Similarly, significant ARID1A binding to the *AURKA* promoter was observed in RKO cells expressing wild-type *ARID1A* (Fig. 5k), and RNA-Pol II binding to the *AURKA* promoter was significantly lower in these cells than in the parental RKO cells (Fig. 5l). ARID1A is known to play a key role in targeting the SWI/SNF complex to DNA via its ARID-DNA-binding domain[44]. We thus wonder whether other core components of the SWI/SNF complex are recruited to the *AURKA* promoter for transcription regulation and, if so, whether the recruitment is dependent on ARID1A. Hence, we next performed ChIP assays of the AURKA promoter using antibodies against the two core components of SWI/SNF, BRG1 (*SMARCA4*) and SNF5 (*SMARCB1*), in *ARID1A*$^{+/+}$ and *ARID1A*$^{-/-}$ HCT116 cells. As shown by the ChIP results, the two core components were indeed recruited to the *AURKA* promoter in *ARID1A*$^{+/+}$ cells but not in *ARID1A*$^{-/-}$ cells (Fig. 5h, i). These data suggest that the SWI/SNF complex is recruited to the *AURKA* promoter via ARID1A-dependent targeting and that it represses *AURKA* transcription in CRC cells. Next, to analyze the generality of the negative AURKA regulation by ARID1A in CRC, we measured AURKA and ARID1A expression levels in six CRC cell lines with different genetic backgrounds. We observed a clear inverse correlation between ARID1A and AURKA levels in the six CRC cell lines (Fig. 5m). In agreement with this finding, a negative correlation between the gene expression levels of ARID1A and AURKA was observed in a large number of CRC patient samples (Supplementary Fig. 6c), further demonstrating the negative regulation of AURKA expression by ARID1A in CRC cells.

**ARID1A deficiency results in persistent CDC25C activation.** When overexpressed in cells, AURKA functions as an oncogene by overriding cell cycle checkpoints[45], enhancing cell proliferation[46], and suppressing apoptotic pathways[47,48]. The overexpression of an oncogene often causes cells to become addicted to oncogenic signaling for their growth and survival, a phenomenon called oncogene addiction[49]. Since AURKA was overexpressed in ARID1A-deficient cells, we analyzed AURKA downstream effectors that are critical for cell cycle progression and proliferation. AURKA is known to phosphorylate CDC25C at Ser198 via PLK1 and activate CDC25C nuclear translocation[24]. When active, the phosphatase activity of CDC25C dephosphorylates CDC2 at Tyr15 and promotes cellular G2/M transition[50]. CDC25C phosphorylation at Ser198 was significantly increased in parallel with decreased CDC2 phosphorylation at Tyr15 in all three *ARID1A*$^{-/-}$ clones (Fig. 6a). Treating cells with either *AURKA* siRNA or small-molecule AURKAi significantly decreased CDC25C phosphorylation at Ser198 and increased CDC2 phosphorylation at Tyr15, especially in *ARID1A*$^{-/-}$ cells (Fig. 6b, c). Likewise, cytoplasmic CDC25C observed in ARID1A wild-type cells was largely translocated into the nucleus in *ARID1A*$^{-/-}$ cells, suggesting the persistent activation of CDC25C (Fig. 6d). *AURKA* silencing significantly reversed the nuclear localization of CDC25C in *ARID1A*$^{-/-}$ cells (Fig. 6d). AURKAi treatment also had a similar effect on CDC25C localization (Supplementary Fig. 7). To further test whether the AURKA–PLK1–CDC25C–CDC2 axis is a target pathway of synthetic lethality in *ARID1A*$^{-/-}$ cells, we examined the effects of the genetic or pharmacological inhibition of PLK1 and CDC25C on viability in *ARID1A* isogenic cells. A small-molecule inhibitor of CDC25C, as well as *PLK1* siRNA, selectively inhibited the growth of *ARID1A*$^{-/-}$ cells (Fig. 6e, f). Together, these results suggest that ARID1A-deficient cells have increased AURKA expression and persistent activation of CDC25C, a key factor in G2/M transition, rendering the cells addicted to this pathway. This oncogene addiction is likely to facilitate the selective vulnerability induced by inhibitors of AURKA or its downstream effectors in *ARID1A*$^{-/-}$ cells (Fig. 6g).

## Discussion

The SWI/SNF chromatin remodeling complex mediates diverse biological pathways by epigenetically regulating gene expression[1]. ARID1A, a component of the SWI/SNF complex, contains a DNA-binding (ARID) domain and plays a crucial role in targeting the complex to gene promoters[44]. By changing the position of nucleosomes, ARID1A and the SWI/SNF complex are capable of activating or repressing the transcription of hundreds of target genes[51], including p21 (*CDKN1A*)[52], SMAD3[52], thrombospondin 1 (*THBS1*)[44], and telomerase reverse transcriptase (*TERT*)[53]. HDAC6 has recently been identified as a target gene of ARID1A for transcription repression in OCCC; in fact, OCCC cells with ARID1A loss were shown to be sensitized to HDAC6 inhibition[19].

In the present study, we show that *AURKA* is a target gene of ARID1A for transcription repression, and it interacts functionally with ARID1A in a synthetic lethal manner in CRC cells. Small-molecule inhibitors or siRNA silencing of *AURKA* caused strong synthetic lethality in ARID1A-deficient CRC cells. The synthetic lethality of ARID1A and AURKA was verified in three different *ARID1A* isogenic CRC pairs and three ovarian cancer cell lines with different ARID1A statuses; however, the synthetic lethality in ovarian cancer cells needs to be further investigated. Phenotypically, the inhibition of AURKA induced cellular G2/M arrest in both *ARID1A*$^{+/+}$ and *ARID1A*$^{-/-}$ cells, but it generated considerable multinucleated *ARID1A*$^{-/-}$ cells, leading to the preferential induction of apoptosis. This phenotype is in agreement with previous observations that AURKA inhibition significantly delayed the cell cycle at the G2/M phase, followed by the induction of endoreduplication, aneuploidy, and apoptosis, depending on the cell type[54,55]. ARID1A is known to interact with ATR and contribute to cellular DNA damage repair and G2/M checkpoint activity[17]. The lack of ARID1A activity allows cells with DNA damage to pass through the G2/M checkpoint, which potentially causes genomic instability[17]. AURKA inhibition impairs the spindle assembly checkpoint and induces abnormal

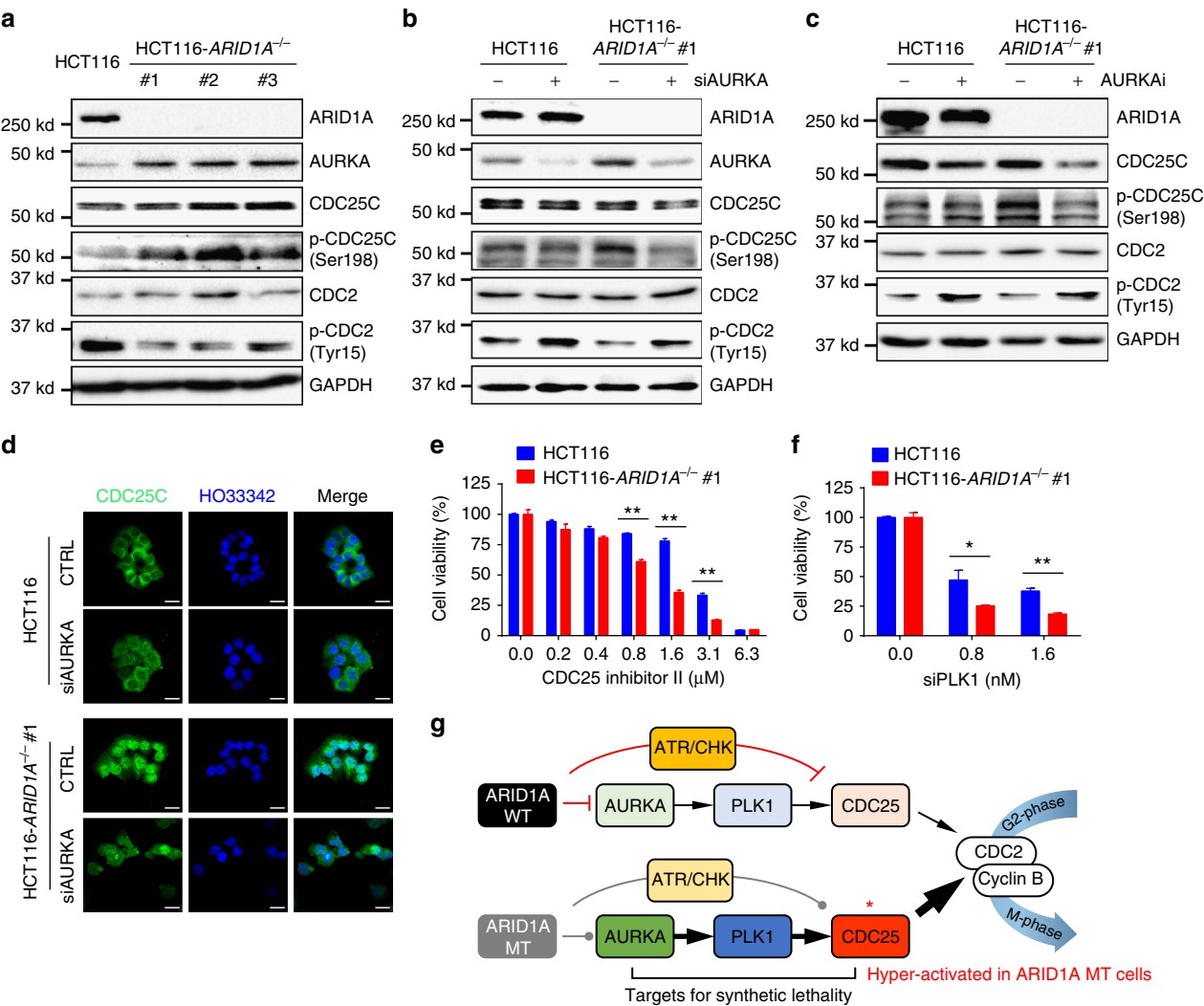

**Fig. 6** AURKA–CDC25C axis is a target for synthetic lethality in ARID1A-KO colorectal cancer cells. **a** Immunoblots showing upregulation of AURKA and phosphorylated CDC25C at Ser198 levels and downregulation of phosphorylated CDC2 at Tyr15 level in *ARID1A*−/− HCT116 cells. **b**, **c** Inhibition of CDC25C phosphorylation at Ser198 and increased phosphorylation of CDC2 at Tyr15 by *AURKA* silencing (**b**) and AURKAi treatment (**c**). **d** Immunofluorescence analysis of CDC25C localization in *ARID1A* isogenic cells treated with or without *AURKA* siRNA. Scale bars, 20 μm. **e** Synthetic lethality in *ARID1A*−/− HCT116 cells by CDC25 inhibitor II. **f** Synthetic lethality in *ARID1A*−/− HCT116 cells by *PLK1* siRNA (siPLK1). Error bars represent s.d. *$P < 0.05$; **$P < 0.01$, Student's *t* test. **g** Working model of the synthetic lethality between ARID1A and AURKA. In *ARID1A* wild-type (WT) cells, AURKA expression is negatively regulated by ARID1A, thereby reducing the activity of AURKA downstream pathway, including PLK1 and CDC25C. In ARID1A mutant (MT) cells, AURKA-PLK1-CDC25C pathway is upregulated. In addition, CDC25C activity is negatively regulated by ARID1A–ATR-CHK (checkpoint kinase) pathway under DNA damage conditions, thereby strictly controlling the CDC25 activity in *ARID1A* WT cells. In ARID1A MT cells, ARID1A-ATR-CHK pathways is impaired and thus CDC25C activity is de-repressed, causing it in hyper-active state. In this condition, cells can be addicted to AURKA–CDC25C pathway for cell survival and proliferation. Therefore, AURKA–CDC25C axis becomes a target for synthetic lethality in ARID1A-deficient cells

chromatin segregation to cause chromosomal instability[55]. The combination of ARID1A deficiency and AURKA inhibition could cause severe genomic and chromosomal abnormalities, which are intolerable to cells and lead to apoptosis.

Mechanistically, ARID1A loss in CRC cells enhanced *AURKA* transcription, making the cells addicted to AURKA signaling for their growth and survival. Cellular oncogene addiction due to high AURKA expression was evidenced by the persistent activation of CDC25C, a downstream effector of AURKA, in *ARID1A*−/− cells; in these cells, the pharmacological or genetic perturbation of AURKA was able to reverse the CDC25C activation. Inhibiting PLK1, a direct upstream target of CDC25C, or CDC25C itself recapitulated the synthetic lethality phenotype in

ARID1A-deficient cells, further supporting the idea that the AURKA–PLK1–CDC25C axis is a key pathway activated in ARID1A-deficient CRC and that it could serve as a therapeutic target. CDC25C is also downstream of the DNA damage repair pathway in which ATR-activated checkpoint kinase 1 (CHK1) phosphorylates and inhibits CDC25C activity[56]. Cells lacking ARID1A exhibited impaired ATR and CHK1 activation upon DNA damage[17]. Therefore, ARID1A loss could also contribute to the persistent activation of CDC25C through inactivating the ATR/CHK pathway. Therefore, it can be postulated that ARID1A-deficient cells have persistent CDC25C activation through AURKA-mediated signal activation and the lack of ATR/CHK-mediated inhibitory signaling (Fig. 6g).

Our findings are further strengthened by recent two reports showing that two other components of the SWI/SNF complex are involved in the downregulation of AURKA expression and cancer cell sensitivity to AURKA inhibitors. Lee et al.[57] reported that SNF5, a component of SWI/SNF, represses *AURKA* transcription in rhabdoid tumors. AURKA is overexpressed in SNF5 mutant rhabdoid tumors, and *AURKA* silencing sensitized the tumor cells to apoptosis induction[57]. More recently, Tagal et al.[58] showed that AURKA is essential for the survival of non-small cell lung cancer (NSCLC) cells that harbor inactivation mutations in BRG1, another SWI/SNF component protein. However, it is unclear whether each component of the SWI/SNF complex causes the synthetic lethality independently or whether they work as a complex. Our ChIP analysis of the *AURKA* promoter with antibodies against the two core components in *ARID1A*$^{+/+}$ and *ARID1A*$^{-/-}$ cells demonstrated that SNF5 and BRG1 targeting to the *AURKA* promoter is dependent on ARID1A. ARID1A contains a DNA-binding (ARID) domain and is known to play a key role in targeting the complex to the target gene promoter[44]. On the other hand, SNF5 is essential for the formation of the SWI/SNF complex[59], and BRG1 provides energy derived from ATP hydrolysis to the complex for the nucleosome remodeling activity[60]. Given the essentiality of the three components in nucleosome remodeling and the transcription regulation functions of the SWI/SNF complex, it is apparent that the entire SWI/SNF complex has a synthetic lethality interaction with AURKA in tumor cells in which mutations in the key components of the SWI/SNF complex causes the induced essentiality or the oncogene addiction of AURKA for cell survival. The observed synthetic lethality in CRC and ovarian cancer models, together with the reported synthetic lethality interactions between other SWI/SNF components and AURKA in rhabdoid tumors and NSCLC models, clearly indicate the potentially broad relevance of our findings to other cancer types where defective SWI/SNF components exist. Indeed, at least five components of the SWI/SNF complex, including SNF5, BAF180, ARID1A, BRG1, and BRD7, have been reported to be frequently mutated in a variety of tu~mor types, such as familial schwannomatosis (30–40% mutation frequency in SNF5), small-cell hepatoblastomas (36% mutation frequency in SNF5), epitheliod sarcomas (55% mutation frequency in SNF5), renal cell carcinoma (41% mutation frequency in BAF180), endometriod carcinoma (35% mutation frequency in ARID1A), and medulloblastoma (3% mutation frequency in ARID1A and BRG1), in addition to CRC, NSCLC, ovarian, and rhabdoid tumors (reviewed by Wilson and Roberts)[1].

The HCT116 CRC model used in this study has a BRG1 point mutation (L1149P)[61] and a *PIK3CA* hotspot mutation (H1047R)[35]. Since EZH2-ARID1A synthetic lethality is mediated by PIK3IP1, which is an endogenous inhibitor of PIK3CA, the *PIK3CA* activating mutation in HCT116 cells may be another possible contributor to the synthetic lethality of the EZH2 inhibitor, in addition to the lack of PIK3IP1 expression regulation by ARID1A in this model. The BRG1 mutation (L1149P) in HCT116 cells has been well characterized previously[61,62]. This mutation does not affect SWI/SNF complex formation, and the BRG1 mutant complexes remain functional in the presence of BRM, another SWI/SNF component that is homologous and partially redundant to BRG1[62]. Based on the findings of our group and others, BRG1 mutations in HCT116 cells do not affect the synthetic lethality interaction of ARID1A and AURKA or other known targets, including PARP1[17] and ATR[18]. However, the BRM compensation for BRG1 deficiency may occur in a gene-specific manner as they have different promoter preferences[63]. Therefore, it cannot be ruled out that BRG1 mutations could potentially affect the synthetic lethality interaction of ARID1A and other targets in the HCT116 model.

AURKA has been increasingly recognized as a target for cancer therapy due to its high expression in many cancer types. Several small-molecule kinase inhibitors, such as alisertib, danusertib, MK-5108, and ENMD-2076, that target AURKA have entered clinical trials for cancer treatment (https://clinicaltrials.gov/). Alisertib (MLN8237) is the most clinically advanced AURKA inhibitor, and it is currently under phase I/II/III clinical investigation for treating leukemia and many other solid tumors[64]. The clinical efficacy of alisertib vary depending on the tumor type, and some cases of serious side effects have been described[65]. However, the potential clinical effect of alisertib is promising as it improved progression-free survival and the duration of disease stability for various tumor types, and the reported side effects were manageable in many cases[66]. To date, clinical studies of AURKA inhibitors in hematologic malignancies have moved fast, but there has been slow progress in solid tumor studies. Therefore, prompt clinical investigations of AURKA inhibitors for treating solid tumors with ARID1A or SWI/SNF complex deficiency where AURKA is highly expressed are warranted. In summary, our data demonstrate that *AURKA* is a target gene of the ARID1A-containing SWI/SNF nucleosome remodeling complex and is a target for inducing synthetic lethality in ARID1A-deficient CRC cells. Persistent CDC25C activation in ARID1A-deficient cells and its inhibition with AURKA inhibitors suggest that the AURKA–CDC25C axis could be a promising therapeutic target for treating CRC without ARID1A expression.

## Methods

**Cell culture and reagents**. CRC cell lines, HCT116, RKO, SW480, LOVO, HCT15, and DLD1, were obtained from American Type Culture Collection (ATCC, Manassas, VA), which have been authenticated by the provider. These cell lines are not listed in the International Cell Line Authentication Committee (ICLAC)'s misidentified cell lines database (http://iclac.org/). HCT116, SW480, HCT15, and DLD1 cells were cultured in RPMI-1640 media supplemented with 10% fetal bovine serum (FBS) and 1% penicillin/streptomycin. RKO and LOVO cells were cultured in Dulbecco's modified Eagle's medium supplemented with 10% FBS and 1% penicillin/streptomycin. Cells were maintained in a humidified incubator adjusted with 5% $CO_2$ at 37 °C. All the cell lines were regularly examined for mycoplasma contamination by the iPSC Core facility in the Faculty of Health Sciences, University of Macau (https://fhs.umac.mo/research/ipsc-core/). CDC25 Phosphatase inhibitor II (sc-202987) was purchased from Santa Cruz Biotechnology (Dallas, TX) and Aurora A inhibitor I (S1451) and tubastatin A (S8049) were from Selleck Chemicals (Houston, TX). VE-821 and olaparib were kindly gifted by Dr. Chuxia Deng and EPZ-6438 was from Dr. Gang Li (Faculty of Health Sciences, University of Macau, Macau SAR, China).

**Generation of *ARID1A*$^{-/-}$ cells**. *ARID1A* KO constructs, including CRISPR-Cas9 plasmid, a pool of three sgRNAs targeting the exon 2 (5'-GCGGTACCCGAT GACCATGC-3' and 5'-ATGGTCATCGGGTACCGCTG-3') and exon 4 (5'-CCC CTCAATGACCTCCAGTA-3') of ARID1A and HDR donor plasmid containing RFP, and puromycin-resistant gene were purchased from Santa Cruz Biotechnology. HCT116 cells were transfected with the *ARID1A* KO constructs using the Lipofectamine 3000 (Thermo Fisher Scientific, Waltham, MA). The *ARID1A*$^{-/-}$ clones were selected with RFP fluorescence and puromycin (1 μg ml$^{-1}$). The *ARID1A*$^{-/-}$ was verified with immunoblot and Sanger sequencing of the genomic *ARID1A* locus containing HDR insertion or indel mutations. The sequencing primers were designed to amplify the *ARID1A* exons 2 and 4 containing the 3 sgRNA target sites. The primer sequence is as follows: (exon-2 forward) 5'-TGG ATCAGATGGGCAAGATG-3'; (exon-2 reverse) 5'-GCCAGTCAGGTCAA GAGAAA-3'; (exon-4 forward) 5'-GAGACAGTCCCATAACCCTTTC-3'; and (exon-4 reverse) 5'-AGGGAGACAGAACAGACATCTA-3'. The target region was amplified and the sequencing results are shown in Supplementary Fig. 1d.

**Lentiviral overexpression of ARID1A**. pLenti-puro-ARID1A was a gift from Ie-Ming Shih (Addgene plasmid #39478)[52]. pMDLg/pRRE, pRSV-Rev, and pMD2.G were gifts from Didier Trono (Addgene plasmids #12251, #12253, and #12259)[67]. The lentiviral plasmids were transfected to 293FT cells with Lipofectamine 3000. The supernatant was collected and the lentiviral particles were concentrated using a Lenti-X Concentrator (Clontech, Mountain View, CA). RKO cells were transduced with the concentrated lentiviral particles and the ARID1A-overexpressing clones were selected with 1 μg ml$^{-1}$ puromycin. The ARID1A overexpression was verified with immunoblot.

**Epigenetic drug screening and cell viability measurement**. Epigenetics Compound Library (L1900) containing 128 small-molecule inhibitors of epigenetics proteins was purchased from Selleck Chemicals. Each compound was arrayed in 384-well plates in an 8-dose, inter-plate titration format, ranging from 14 nM to 30 μM of final concentrations. HCT116 $ARID1A^{+/+}$ or $ARID1A^{-/-}$ #1 cells were seeded at 2000 cells per well in the 384-well plates containing working dilution of the compound library and incubated for 72 h at 37 °C $CO_2$ incubator. All the liquid handling was done with Liquidator-96 multi-well pipettor (Mettler Toledo, Columbus, OH). For cell viability measurement, cells were incubated with AlamarBlue solution (Sigma-Aldrich, St. Louis, MO) at 10% for 3 h and the fluorescence signal (ex560/em590) at the bottom of the plate was measured with SpectraMax-M5 (Molecular Devices, Sunnyvale, CA). The individual $IC_{50}$ values of each compound against the $ARID1A$ isogenic cell pair were calculated with GraphPad Prism 6.0 (GraphPad Software, La Jolla, CA). The screening was done in duplicated and the average $IC_{50}$ value from the two screenings were used to identify synthetic lethality hits. SI was calculated according to the following equation: $SI = IC_{50}^{ARID1A(+/+)}/IC_{50}^{ARID1A(-/-)}$. Compounds with SI > 2 were selected as synthetic lethality candidates.

**siRNA silencing of AURKA**. Two different $AURKA$ siRNA (siAURKA#1 and siAURKA#2) and a $PLK1$ siRNA were designed and synthesized from Integrated DNA Technologies (Coralville, IA) with following sequences: siAURKA#1, 5'-CU CUAUAAACUGUUCCAAGUGGUGCAU-3', siAURKA#2, 5'- GCACAAUU CUCGUGGCUACUUUCACUU-3', and siPLK1, 5'-GUACUAUUAAGAGGA GACUUGAAAA-3'. The siRNA transfection was performed with Lipofectamine-3000 according to the manufacturer's instructions. Reverse transfection method was used for the silencing of $AURKA$ in a 96-well plate. Briefly, in each well, 5 μL of siAURKA at an indicated concentration suspended in serum-free medium was mixed with 5 μL serum-free medium containing 0.1 μL Lipofectamine 3000 and the transfection mixture was incubated at room temperature for 20 min. Cells were trypsinized and 100 μL of cell suspension (5000 cells per well) was added to each well containing the transfection mixture. Cells were then incubated for 72 h in a $CO_2$ incubator and cell images were taken with IncuCyte ZOOM (Essen BioScience, Ann Arbor, MI). Integrated cell density was measured with the IncuCyte ZOOM software to assess cell viability. The silencing efficiency was assessed with immunoblot using an anti-AURKA antibody.

**Immunoblot and antibodies**. Whole-cell protein extracts were prepared with ice-cold RIPA buffer (25 mM Tris-HCl pH 7.6, 150 mM NaCl, 1% NP-40, 1% sodium deoxycholate, 0.1% sodium dodecyl sulfate (SDS)) with Complete Protease Inhibitor Cocktail (Roche Life Sciences, Indianapolis, IN). Each aliquot of protein sample was run on a SDS-polyacrylamide gel electrophoresis and transferred onto a nitrocellulose membrane for immunoblotting with primary antibodies, including ARID1A (Santa Cruz, sc-32761, 1:1000 dilution), CDC25C (Santa Cruz, sc-327, 1:2000 dilution), CDC2 (Santa Cruz, sc-54, 1:2000 dilution), GAPDH (Santa Cruz, sc-365062, 1:5000 dilution), HSP90 (Santa Cruz, sc-69703, 1:5000 dilution), α-tubulin (Santa Cruz, sc-5286, 1:5000 dilution), p-CDC25C (Cell Signaling, #9529s, 1:1000 dilution), p-CDC2 (Cell Signaling, #9111s, 1:1000 dilution), AURKA (Cell Signaling, #14475s, 1:2000 dilution), and cleaved caspase 3 (Cell Signaling, #9661s, 1:2000 dilution) antibodies, followed by horseradish peroxidase-conjugated secondary antibodies (Santa Cruz). Uncropped versions of all blots are shown in Supplementary Figs. 8-12.

**Tumor xenograft mouse model**. All animal procedures were approved by the Animal Research Ethics Committee of the University of Macau and were carried out according to ARRIVE guidelines[68]. Six-week-old, female BALB/c nude mice were implanted with $ARID1A^{+/+}$ (left flank) and $ARID1A^{-/-}$ (right flank) HCT116 cells suspended in Matrigel. When both tumors were palpable, mice were randomized into 3 groups ($n = 5$ mice per group) of equal tumor volume for treatment with vehicle and AURKAi, but the researchers were not blinded to the groups when performing the experiments. Mice were treated with vehicle (sterile saline containing 5% dimethyl sulfoxide, 5% tween-80, and 5% polyethylene glycol-400) or AURKAi (30 and 60 mg kg$^{-1}$, daily) via i.p. injection for 24 days. The sample size ($n = 5$ per group) in the animal experiment was chosen according to the report by Charan and Kantharia[69]. The similar procedure of xenograft experiments were conducted with RKO-$ARID1A$ isogenic cell pair ($n = 6$ mice per group). The tumor size was periodically measured with a Vernier caliper and the tumor volume was calculated based on the modified ellipsoid formula (long axis × short axis$^2$ × π/6). At the end of experiments, mice were sacrificed and the tumors were harvested for weighing and further analyses. Mice body weights were measured regularly during the drug injection period to assess potential drug toxicity.

**Immunofluorescence analyses**. Cells were seeded on a Nunc Lab-Tek II 8-Chamber Slide (Thermo Fisher Scientific) and treated with compound or siRNA for the indicated time. Cells were then fixed with 4% paraformaldehyde and permeabilized with 0.5% Triton X-100, followed by blocking with 2% bovine serum albumin for 1 h. Cells were incubated with primary antibodies, including AURKA (Cell Signaling, #14475s, 1:100 dilution), CDC25C (Santa Cruz, sc-327, 1:100 dilution), and α-tubulin (Santa Cruz, sc-5286, 1:100 dilution) in the blocking buffer

overnight at 4 °C, followed by the incubation with secondary antibodies conjugated with Alexa Fluor 488 or Alexa Fluor 647 for 1 h at room temperature. The nuclei were stained with Hoechst 33342 (Thermo Fisher Scientific). After washing with phosphate-buffered saline (PBS), cells were mounted with Immu-mount (Thermo Fisher Scientific) and observed under a Carl Zeiss LSM 710 confocal microscope (Carl Zeiss, Thornwood, NY).

**ChIP of AURKA promoter**. ChIP was performed with Imprint Chromatin Immunoprecipitation Kit (CHP1, Sigma-Aldrich) as per the manufacturer's instructions. Briefly, cells were treated with 1% formaldehyde to crosslink DNA and proteins prior to prepare nuclear fraction. The nuclear pellet was sonicated with a Bioruptor Sonication System (Diagenode, Denville, NJ) to shear total DNA. Then the samples were immunoprecipitated with anti-ARID1A (Santa Cruz, sc-32761, 10 μg per sample for ChIP), anti-RNA polymerase II (Sigma-Aldrich, CHP1, 1 μg per sample for ChIP), anti-BRG1 (Santa Cruz, sc-17796×, 4 μg per sample for ChIP), and anti-SNF5 (Santa Cruz, sc-166165×, 4 μg per sample for ChIP) ChIP-grade antibodies using the assay wells pre-coated with protein A. A normal mouse IgG was used as a non-specific antibody control for immunoprecipitation. After reverse crosslinking and protein degradation of the samples, the ChIP DNA were analyzed with qPCR against the $AURKA$ promoter region. Following are sequences of the primer pairs used to analyze ChIP DNA: Primer #1 ($-1$ to $-75$ bp from the $AURKA$ TSS): forward primer 5'-ACAGGTCTGGCTGGCCGTTGGC-3' and reverse primer 5'- GGCGCA CACCGCGCGCAGGCG-3'; Primer #2 ($-3892$ to $-3787$ bp from the $AURKA$ TSS): forward primer 5'-AGGACGACTAGGTGGTAGATAAA-3' and reverse primer 5'-GTACTTGCATCCTCAGCAGAA-3'; Primer #3 ($+3201$ to $+3296$ bp from the $AURKA$ TSS): forward primer 5'-CAAGGTCCTGATCCTTACTC AAC-3' and reverse primer 5'-CCTTTATCATTTGGGCTGTTTCC-3' (Integrated DNA Technologies); and Control primer (ORF-free region): forward primer 5'-CC TGGAGGGCTTGGAGATG-3' and reverse primer 5'-GATCCTACGGC TGGCTGTGA-3' (a kind gift of Professor Edwin Cheung at the University of Macau)[70]. The ChIP-qPCR data were expressed as fold enrichment (ΔΔCt method) according to the manufacturer's instruction. Briefly, each ChIP DNA fraction's Ct value was normalized to the input DNA fraction Ct value (ΔCt [normalized ChIP]). The ΔCt for antibody ChIP was normalized to the IgG control ChIP ΔCt (ΔΔCt [antibody ChIP/IgG ChIP]). Fold enrichment of the specific site was calculated according to the following equation: fold enrichment = $2^{(-ΔΔCt\ [antibody\ ChIP/IgG\ ChIP])}$.

**Cell cycle and apoptosis assays**. Cell cycle and apoptosis were analyzed with a standard flow cytometry protocol. Briefly, for cell cycle analysis, cells were harvested and then fixed with 70% ethanol overnight at $-20$ °C prior to propidium iodide (PI) staining. For apoptosis analysis, FITC–Annexin V Apoptosis Detection Kit with PI (BioLegend, San Diego, CA) was used. Briefly, cells were washed with ice-cold PBS and re-suspended with Annexin V staining buffer. The cells were then stained with fluorescein isothiocyanate (FITC)–Annxin V and PI at room temperature for 15 min and then analyzed immediately with a BD Accuri C6 flow cytometer (BD Biosciences, San Jose, CA).

**RT and qPCR**. Total cellular RNA was harvested with the RNeasy Mini Kit, followed by on-column DNAse digestion (Qiagen, Hilden, Germany). RT was performed with the High-Capacity cDNA Reverse Transcription Kit (Thermo Fisher Scientific). AURKA transcription level was determined with SYBR Green Supermix (Bio-Rad, Hercules, CA) with the primer sequences as: forward primer 5'-CAG TACATGCTCCATCTTCCAG-3' and reverse primer 5'-AAA GAACTCCAAGGCTCCAG-3' (Integrated DNA Technologies).

**Statistical analysis**. All data were expressed as the mean ± standard deviation (s.d.). Statistical significance of differences between control and test groups was determined by Student's $t$ test or one-sample $t$ test using Graphpad Prism 6 (GraphPad Software, La Jolla, CA). Statistical analysis of differences between two dose–response curves was determined by analysis of variance using Graphpad Prism 6. All statistical tests were two tailed. $P$ values <0.05 were considered significant.

**Data availability**. All relevant data that are not present in the paper or Supplementary Data are available from the authors.

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

## Acknowledgements

This study was supported by the Multi-Year Research Grant of the University of Macau (MYRG2017-00176-FHS and MYRG2015-00181-FHS to J.S.S.) and the Science and Technology Development Fund (FDCT) of Macau SAR (FDCT/024/2015/A1 to J.S.S.).

## Author contributions

C.W. performed experiments for gene editing and cell line generation. C.W. and E.J.Y. conducted drug screening and analyzed data. C.W., J.L., Y.L., and B.Z. conducted in vitro cell-based experiments and animal studies. J.S.S. designed the research and supervised the project. C.W. and J.S.S. wrote the manuscript. All the authors reviewed the manuscript.

## Additional information

**Competing interests:** The authors declare no competing interests.

