## [Peer Review File · Nature Communications]

Reviewers' comments:

Reviewer #1 (Remarks to the Author):

By screening a drug library using isogenic WT and Arid1a-KO HCT116 cells, the authors identify a synthetic lethal effect between AURKA and Arid1a. Specifically, Arid1a-KO cells are more sensitive to AURKA inhibition. This is elegant, well written, organized, and important. There are a few small points that can be addressed to strengthen the paper.

Major points

- The authors propose a model that Arid1a KO induces hyper activated cell cycle proteins by upregulating AURKA expression. A missing piece of information here is whether or not the authors saw a growth/proliferation difference between WT and Arid1a-KO cells. Could the sensitivity to Aurki be due to the fact that any cell with higher proliferative rates is more sensitive to this kind of cell cycle inhibition?

- Most or many Arid1a mutant cancer cells have heterozygous mutations in Arid1a. Is there evidence that in cells with heterozygous mutations in Arid1a that aurki is still effective?

- Finally, is the aurki effective in additional tissue types or cancer cell lines with arid1a mutations? I know there is evidence from RKO cells, but it would be nice if 2-3 other cell lines were tested.

Minor points

1. In Fig.1f, statistics are needed. The authors should add the dose response curves of the other two Arid1a-KO single clones. It is not clear if Fig.1f is generated using 384 well plate. The authors could use 96 well plates by using multiple wells for one data point to minimize the variations of cell viability at high concentrations of AURKAI.

2. In Fig.2a, the authors indicate in the figure legend that 1 μ M of AURKAI was used and the viability of Arid1a-KO cell decreased to half compared with the control cell. But from Fig.1f, this concentration cannot induce an obvious cell death. The authors should explain the discrepancy.

3. In Fig. 2c, the authors should show the western blot data of AURKA treated with different concentrations of siAURKA, to show that AURKA is indeed KD, as we can see from Fig. 2b, the efficiency of KD is relatively low.

4. In Fig. 3d, the authors should include dose response curves (like fig. 1f) for RKO WT and Arid1a-expressing clones #2/3 to support the idea.

5. In Fig. 5e, it would be better if the author include the AURKA mRNA expression data of RKO WT and Arid1a-expressing isogenic pairs to see if AURKA mRNA is down regulated by Arid1a expression.

6. In Fig. 5g, the data cannot support the claim that 'Arid1a may directly repress the AURKA transcription by occupying its promoter'. The authors should include more controls in ChIP-qPCR. Control Primers that away from AURKA promoter region should be used to show that Arid1a indeed bind to the promoter region. In addition, the author should check if it is the case in RKO and RKO-Arid1a cells.

Reviewer #2 (Remarks to the Author):

This study demonstrates that loss of ARID1A expression in 2 human colorectal carcinoma cell lines leads to increased expression of aurora kinase A (AURKA). This increased expression results in a dependency upon a constitutively active G2/M checkpoint pathway for cell viability. Their data demonstrate that these ARID1A-deficient cell lines are sensitive to AURKA inhibitors or AURKA knockdown by siRNA in cell culture and in xenograft models. They also show that ARID1A directly

regulates transcription of the AURKA gene. Finally, they demonstrate the constitutive activation of the AURKA downstream targets PLK1 and CDC25, whose inhibitors also cause apoptosis in the ARID1A-deficient cell lines.

While the authors present generally clear results from a set of well-designed that support their model of ARID1A-AURKA synthetic lethality, the novelty seems somewhat limited. For example, Lee et al. reported in 2011 that loss of another core SWI/SNF complex component, SNF5, resulted in AURKA overexpression concomitant with sensitivity to its knockdown (Cancer Res. 2011 May 1;71(9):3225-35). A more recent report from Tagal et al. showed the same paradigm for sensitivity to AURKA inhibitors in non-small cell lung carcinomas that had lost expression of SMARCA4, one of the ATPases that fuels the chromatin remodeling activity of the complex (Nat Commun. 2017 Jan 19;8:14098). The authors do not cite or discuss either of these reports. The authors also correctly cite that previous reports have identified at least 4 other genes that act as synthetic lethal targets in ARID1A-deficient tumors. However, this manuscript does not compare the efficacy of these established targets with that of AURKA. They also do not discuss the status of AURKAi in clinical trials or whether it offers a better choice for targeted therapy than the inhibitors of the other genes. Therefore, this study would appear more appropriate for publication in a specialized journal for cancer therapeutics.

Major Comments:

1) Did the authors observe any of the other previously-identified epigenetic modifiers that act as synthetic lethals for ARID1A loss in their screen in Figure 1D? If so, how did they compare to the AURKA inhibitors? If not, how do they explain their absence?

2) The authors propose that ARID1A acts as a major regulator of AURKA expression. They should show the levels of AURKA protein for the cell lines in Figure 3a as a further test of that model.

3) They should assess the binding of at least 2 other core members of the SWI/SNF complex by ChIP as shown for ARID1A in Figure 5g. In other words, is the SWI/SNF complex present at the TSS of AURKA in the absence of ARID1A or does it recruit the complex?

Author responses to Referees

Reviewer #1 (Remarks to the Author):

By screening a drug library using isogenic WT and Arid1a-KO HCT116 cells, the authors identify a synthetic lethal effect between AURKA and Arid1a. Specifically, Arid1a-KO cells are more sensitive to AURKA inhibition. This is elegant, well written, organized, and important. There are a few small points that can be addressed to strengthen the paper.

Response: We are grateful to the reviewer for thoughtful and constructive comments to improve the quality of our paper. In our revised version, we have included several additional data to address the reviewer's concerns. Described below are point-to-point responses to the reviewer's specific comments.

Major points

- The authors propose a model that Arid1a KO induces hyper activated cell cycle proteins by upregulating AURKA expression. A missing piece of information here is whether or not the authors saw a growth/proliferation difference between WT and Arid1a-KO cells. Could the sensitivity to Aurki be due to the fact that any cell with higher proliferative rates is more sensitive to this kind of cell cycle inhibition?

Response: We agree with the reviewer's comment that the synthetic lethality effect might be due to the differential growth rate between the two cell lines. Therefore, we examined the growth rates of HCT116 WT/ARID1A-KO and RKO WT/ARID1A-deficient cells using an IncuCyte ZOOM real-time cell imager. As shown in the figures below (Supplementary Fig. 2a, b), the growth rates between WT and ARID1A-deficient CRC cells were largely similar.

Supplementary Figure 2a and b. The growth rates of ARID1A-isogenic colorectal cancer cells. ARID1A-isogenic HCT116 (a) and RKO (b) cell lines were grown in a 96-well ImageLock Microplate until confluent and assessed for real-time growth rate with IncuCyte-ZOOM. No significant difference was observed in in vitro growth rate between ARID1A-WT and KO cells.

In fact, several recent papers also demonstrated that the loss of ARID1A does not enhance cell proliferation in short-term cell culture (Bitler et al., 2015, Nat Med; Sun et al, 2016, Cell Stem

Cell; Bitler et al., 2017, Nat Cell Biol). Based on these reports, the ARID1A loss is likely to make cells bypass the growth arrest under the cell/tissue damage conditions and facilitate cell survival, rather than enhance cell proliferation in regular, short-term culture. Therefore, we believe that the synthetic lethality by AURK*Ai* is not due to the differential growth rate between ARID1A-WT and KO cells, but due to the oncogene addiction (or induced essentiality) of AURKA in ARID1A deficient cells as described in our proposed model.

We have included following description in the main text of the revised version:

“... In vitro growth rates between ARID1A-wildtype and KO HCT116 CRC cells were similar in short-term culture (Supplementary Fig. 2a), which was in agreement with previous reports^{18, 21, 38}...”

“... Similarly to the HCT116 ARID1A-isogenic pair, the growth rates of RKO parental and ARID1A-expressing clones were largely similar (Supplementary Fig. 2b)...”

- Most or many *Arid1a* mutant cancer cells have heterozygous mutations in *Arid1a*. Is there evidence that in cells with heterozygous mutations in *Arid1a* that *aurki* is still effective?

Response: We are grateful to the reviewer to point out this important issue. As the reviewer indicated, only 30% of the cancer with ARID1A mutations were both alleles affected in ovarian cancer, suggesting that a large portion of cancer ARID1A mutation are heterozygous mutations (Jones et al, 2010, Science; Wiegand et al, 2010, New Eng J Med; Wu and Roberts, 2013 Cancer Discov). Interestingly more than 70% of those with ARID1A heterozygous mutation have loss of protein expression, suggesting a phenotypic similarity between homozygous and heterozygous mutations (Wiegand et al, 2010, New Eng J Med; Wu and Roberts, 2013 Cancer Discov). From a translational view, it would be important to test the synthetic lethality in CRC cells with ARID1A heterozygous mutation. Therefore, we generated homozygous and heterozygous ARID1A-KO CRC cells using the CRISPR/Cas-9 and tested the efficacy of AURK*Ai*. For detail, please see our responses for the next comment.

- Finally, is the *aurki* effective in additional tissue types or cancer cell lines with *arid1a* mutations? I know there is evidence from RKO cells, but it would be nice if 2-3 other cell lines were tested.

Response: To address this and the previous comments, we used another CRC cells, SW480, to generate both ARID1A heterozygous and homozygous KO clones using a CRISPR/Cas9. As shown in the figure below (Supplementary Fig. 4a, b), we have successfully isolated heterozygous and homozygous ARID1A KO clones from the genome edited SW480. The KO was verified with the genomic PCR analysis of ARID1A in the clones we isolated. The homozygous KO clones B4 and B8 have no detectable ARID1A expression, while the heterozygous KO clone B9 has barely detectable ARID1A expression. We observed further gradual loss of the ARID1A protein level in this heterozygote KO clone (B9) over weeks of culture period (data not shown). Consistent with this observation, previous reports also showed that more than 70% of ARID1A-heterozygous mutation lacked protein expression (Wiegand et al, 2010, New Eng J Med; Wu and Roberts, 2013 Cancer Discov). We then examined the synthetic lethality effect of AURK*Ai* in the SW480 ARID1A-homozygous and -heterozygous KO clones. Indeed, the heterozygous KO as well as the two homozygous KO clones showed greater sensitivity to AURK*Ai* treatment compared to the

parental SW480 cells expressing wildtype ARID1A (Supplementary Fig. 4c). These data suggest that synthetic lethality between ARID1A and AURKA is largely common among CRC cells and that the synthetic lethality is applicable to heterozygous ARID1A mutations with loss of protein expression.

Supplementary Figure 4. Generation of SW480 ARID1A-KO cells and validation of ARID1A-AURKA synthetic lethality. SW480 colorectal cancer cells harboring wildtype ARID1A was transfected with CRISPR/Cas9 plasmid, sgRNAs targeting ARID1A genomic locus, and homologous-directed repair (HDR) donor plasmid containing puromycin-resistance and red fluorescence protein (RFP) genes to generate ARID1A-KO cell lines. The RFP fluorescent cells were picked up and the KO clones were further selected with puromycin. **a**, Western blot analysis of ARID1A expression in SW480 parental and puromycin-selected ARID1A-KO clones. Clones B4 and B8 have no detectable ARID1A expression, while clone B9 has barely detectable ARID1A expression. **b**, The three clones (B4, B8, and B9) were selected for genomic PCR analysis with primer pairs specific for sgRNA target site within ARID1A locus. Clone B9 has both ARID1A wildtype amplicon (WT) and an amplicon with HDR sequences (HDR insertion), suggesting an ARID1A-heterozygous KO clone. B8 and B4 clones only have HDR insertion amplicon, suggesting ARID1A-homozygous KO clones. **c**, Dose response curves of SW480 ARID1A-isogenic cell lines with AURKAi treatment. Both heterozygous and homozygous ARID1A KO clones show significantly increased sensitivity to AURKAi treatment.

Based on the results we obtained, we have included following description in the main text of the revised version:

“...We further examined the synthetic lethality between ARID1A and AURKA in other CRC and ovarian cancer cells with different ARID1A status. SW480 CRC cells expressing wildtype ARID1A were used to generate ARID1A-KO cells using the CRISPR/Cas-9 gene editing. Since a large portion of ARID1A mutation in patients is heterozygous mutation^{3, 15}, we isolated both homozygous (ARID1A^{-/-}) and heterozygous (ARID1A^{+/-}) ARID1A-KO clones from SW480 to test for the synthetic lethality (Supplementary Fig. 4a, b). The heterozygous ARID1A-KO clone B9 had barely detectable ARID1A protein expression, which was in agreement with previous reports that more than 70% of ARID1A-heterozygous mutation lacked protein expression^{3, 15}. This clone together with the two homozygous ARID1A-KO clones showed greater sensitivity to AURKAi treatment compared to the parental SW480 cells expressing wildtype ARID1A (Supplementary Fig. 4c). These data suggest that synthetic lethality between ARID1A and

AURKA is largely common among CRC cells and that the synthetic lethality is applicable to heterozygous ARID1A mutations with loss of protein expression...

We also tested the AURKAi sensitivity in three ovarian cancer cells with different ARID1A status: SKOV3 (ARID1A-deficient) and HO8910 and ES2 (both wildtype ARID1A). Results showed that ARID1A deficient SKOV3 cells were significantly more sensitive to AURKAi treatment compared to ARID1A wildtype HO8910 and ES2 ovarian cancer cells (Supplementary Fig. 5a-c).

Supplementary Figure 5. Validation of ARID1A-AURKA synthetic lethality in ovarian cancer cell lines. **a**, Three ovarian cancer cell lines with different ARID1A status, ES2 (ARID1A-WT), HO8910 (ARID1A-WT), and SKOV3 (ARID1A-MT) were treated with AURKAi for three days and nuclei were stained with Hoechst 33342. **b**, Western blot analysis of ARID1A status in three ovarian cancer cell lines. Actin was used as an internal control. **c**, The cell viability was measured by counting the number of cell nuclei with Image J software.

These results suggest that ARID1A-AURKA synthetic lethality is largely common across the cell types, although the synthetic lethality in ovarian cancer cells need to be further elaborated using isogenic cell pairs (as this study is focused on colorectal cancer, we did not explore deeply into the ovarian cancer cells).

We have included following description in results and in discussion sections in the revised version: "...We further verified the synthetic lethality between ARID1A and AURKA in other CRC and ovarian cancer cells with different ARID1A status..."

"...We also tested the AURKAi sensitivity in three ovarian cancer cells with different ARID1A status. ARID1A deficient SKOV3 cells were significantly more sensitive to AURKAi treatment compared to ARID1A wildtype HO8910 and ES2 ovarian cancer cells (Supplementary Fig. 5a-c)..."

"...The synthetic lethality between ARID1A and AURKA has been verified in 3 different ARID1A-isogenic CRC pairs and 3 ovarian cancer cell lines with different ARID1A status, albeit the synthetic lethality in ovarian cancer cells needs to be further elaborated..."

Minor points

1. In Fig.1f, statistics are needed. The authors should add the dose response curves of the other two Arid1a-KO single clones. It is not clear if Fig.1f is generated using 384 well plate. The authors could use 96 well plates by using multiple wells for one data point to minimize the variations of cell viability at high concentrations of AURKAI.

Response: As the reviewer mentioned, the original dose-response curve (Fig. 1f) was from the two-rounds screening in 384-well plates. Therefore, we re-analyzed the dose-response curve in 96-well plates with freshly prepared AURKAI in all three KO clones (Fig. 1f, in the revised manuscript).

Fig. 1f. Dose response curves of HCT116 ARID1A wildtype and three KO clones treated with Aurora A Inhibitor I (AURKAI) are shown. Error bars represent s.d. (n=9) from three independent experiments. Survival curve of all three KO clones versus wildtype cells P value <0.0001, ANOVA.

2. In Fig.2a, the authors indicate in the figure legend that 1 μM of AURKAI was used and the viability of Arid1a-KO cell decreased to half compared with the control cell. But from Fig.1f, this concentration cannot induce an obvious cell death. The authors should explain the discrepancy.

Response: As described above, the original data (Fig. 1f) were from the 384-well screening. The compound library used for the screening was the commercial library available as DMSO stock solution. The actual concentration in the library stock could be different from the original description as compounds in solution state can be degraded or become unstable over time during freeze/thaw cycles. Therefore, it is very important to validate the results using freshly prepared compound from the powder. Our all follow-up studies were conducted with the freshly prepared compound solution from the powder, thus there was a discrepancy in drug IC50 data between the screening and validation experiments. However, we believe that even drugs are degraded and have higher IC50 than actual, the relative sensitivity difference between cell lines will remain similar (thus we were able to identify hits from screening). To clarify this, and to avoid discrepancy, we re-analyzed the dose-response curve of AURKAI using freshly prepared compound and replaced the Fig. 1f with new data (Fig. 1f, in the revised manuscript).

3. In Fig. 2c, the authors should show the western blot data of AURKA treated with different concentrations of siAURKA, to show that AURKA is indeed KD, as we can see from Fig. 2b, the efficiency of KD is relatively low.

Response: As per the reviewer's suggestion, we used different concentrations of siAURKA in both ARID1A WT and KO cells, and the KD efficiency was again verified with Western blots (Fig. 2b in the revised manuscript).

b

Fig. 2b. Silencing of AURKA expression in ARID1A isogenic cell pair by siRNA (siAURKA). GAPDH was used as a loading control.

4. In Fig. 3d, the authors should include dose response curves (like fig. 1f) for RKO WT and Arid1a-expressing clones #2/3 to support the idea.

Response: As per the reviewer's suggestion, we included dose-response curves for RKO WT and KO clones showing differential sensitivity to AURKAI treatment (Fig. 3b in the revised manuscript)

Fig. 3b. Dose response curves of ARID1A-deficient parental RKO and ARID1A expressing RKO clones treated with AURKAI. Error bars represent s.d. (n=6) from three independent experiments. Survival curve of ARID1A wildtype versus mutant cell lines P value <0.0001, ANOVA.

In addition to AURKAI, we also tested various concentrations of AURKA siRNA in the RKO parental and ARID1A-expressing cells to further verify the synthetic lethality effect by AURKA silencing (Fig. 3c and d in the revised manuscript).

Fig. 3c and d. (c) Synthetic lethality effect of AURKA siRNA (siAURKA) on RKO ARID1A-isogenic pair. Representative cell images were taken with IncuCyte ZOOM. (d) Integrated cell density was measured with IncuCyte ZOOM software as a surrogate for cell viability (*right panels*). ** $P < 0.01$, Student's *t*-test.

5. In Fig. 5e, it would be better if the author include the AURKA mRNA expression data of RKO WT and Arid1a-expressing isogenic pairs to see if AURKA mRNA is down regulated by Arid1a expression.

Response: As per the reviewer's suggestion, we examined RT-qPCR analysis of AURKA mRNA expression in RKO parental and ARID1A-expressing clones. As shown in the results below (Fig. 5j in the revised manuscript), AURKA mRNA was significantly down-regulated in ARID1A-expressing RKO cells.

Fig. 5j. RT-qPCR analysis of AURKA mRNA level in RKO parental and ARID1A expressing clones. * $P < 0.05$ vs RKO (ARID1A-deficient), One sample *t*-test.

6. In Fig. 5g, the data cannot support the claim that 'Arid1a may directly repress the AURKA transcription by occupying its promoter'. The authors should include more controls in ChIP-qPCR. Control Primers that away from AURKA promoter region should be used to show that Arid1a indeed bind to the promoter region. In addition, the author should check if it is the case in RKO and RKO-Arid1a cells.

Response: As we agree with the reviewer's comment about inclusion of additional controls for ChIP experiments to support the claim, we designed two additional sets of primer pairs that target

far down- (+3,201 to +3,296 bp) or up-stream (-3,892 to -3,787 bp) of transcription start site (TSS) of AURKA gene and used them as negative controls for the ARID1A ChIP. These two regions were not enriched by ARID1A ChIP (Supplementary Fig. 6a, b in the revised manuscript).

Supplementary Figure 6. Negative regulation of AURKA expression by ARID1A. a, Target regions of primer pairs used for chromatin immunoprecipitation (ChIP) of AURKA promoter with an anti-ARID1A antibody. b, AURKA promoter ChIP was done with an anti-ARID1A antibody and three indicated primer pairs in HCT116 cells. TSS denotes the transcription start site. * $P < 0.05$, One sample t -test.

We also conducted the ChIP experiments for AURKA promoter using the anti-ARID1A antibody and anti-RNA Pol-II antibody in RKO parental and ARID1A-expressing clones. The results clearly indicated that ARID1A binds to AURKA promoter and represses transcription in RKO ARID1A cells (Fig. 5k, l, in the revised manuscript).

Fig. 5k and l. ChIP of AURKA promoter in ARID1A expressing RKO clone #2 using anti-ARID1A antibody. ** $P < 0.01$ vs IgG, One sample t -test. (l) ChIP of AURKA promoter in RKO parental and ARID1A expressing cells using anti-RNA-Pol II antibody. ** $P < 0.01$ vs RKO (ARID1A-deficient), One sample t -test.

In addition to above described data, we conducted ChIP for other two core components of SWI/SNF complex, BRG1 and SNF5, and found that BRG1 and SNF5 are recruited to AURKA promoter and the recruitment is dependent on ARID1A (Fig. 5h, i, in the revised manuscript). These data further verify that SWI/SNF complex is involved in transcription regulation of AURKA and thus in the synthetic lethality with AURKA.

We again thank the reviewer for all the constructive comments for the manuscript and now believe that our revised manuscript has been substantially improved by addressing the reviewer's concerns.

Reviewer #2 (Remarks to the Author):

This study demonstrates that loss of ARID1A expression in 2 human colorectal carcinoma cell lines leads to increased expression of aurora kinase A (AURKA). This increased expression results in a dependency upon a constitutively active G2/M checkpoint pathway for cell viability. Their data demonstrate that these ARID1A-deficient cell lines are sensitive to AURKA inhibitors or AURKA knockdown by siRNA in cell culture and in xenograft models. They also show that ARID1A directly regulates transcription of the AURKA gene. Finally, they demonstrate the constitutive activation of the AURKA downstream targets PLK1 and CDC25, whose inhibitors also cause apoptosis in the ARID1A-deficient cell lines.

While the authors present generally clear results from a set of well-designed that support their model of ARID1A-AURKA synthetic lethality, the novelty seems somewhat limited. For example, Lee et al. reported in 2011 that loss of another core SWI/SNF complex component, SNF5, resulted in AURKA overexpression concomitant with sensitivity to its knockdown (Cancer Res. 2011 May 1;71(9):3225-35). A more recent report from Tagal et al. showed the same paradigm for sensitivity to AURKA inhibitors in non-small cell lung carcinomas that had lost expression of SMARCA4, one of the ATPases that fuels the chromatin remodeling activity of the complex (Nat Commun. 2017 Jan 19;8:14098). The authors do not cite or discuss either of these reports. The authors also correctly cite that previous reports have identified at least 4 other genes that act as synthetic lethal targets in ARID1A-deficient tumors. However, this manuscript does not compare the efficacy of these established targets with that of AURKA. They also do not discuss the status of AURKAi in clinical trials or whether it offers a better choice for targeted therapy than the inhibitors of the other genes. Therefore, this study would appear more appropriate for publication in a specialized journal for cancer therapeutics.

Response: We greatly appreciate the reviewer for critical and constructive comments for our manuscript, especially for pointing out the recent two key papers we missed from our initial literature study. It was mainly because our initial study focus was on mutations in ARID1A as an independent tumor suppressor in colorectal cancer (CRC) and its synthetic lethality targets without seriously taking it as a part of whole SWI/SNF complex for synthetic lethality, thus missing the reports from the literature screening. As the reviewer indicated, the two recent studies about the synthetic lethality between the two components (SNF5, BRG1) of SWI/SNF complex and AURKA could limit the novelty of our findings that ARID1A, another component of SWI/SNF, has a synthetic lethality with AURKA. However from another view, the two studies may actually

strengthen our current work if we properly discuss their findings and provide further evidence that the two components are functionally linked to ARID1A in the synthetic lethality with AURKA. This will support the idea of the synthetic lethality between entire SWI/SNF5 nucleosome remodeling complex and AURKA. In fact, the two previous reports studied their components as an independent tumor suppressor, like what we did, rather than studied in the context of complex. Therefore the reviewer's suggestions to test whether the two core components can bind to the AURKA promoter in the CRC cells and if so whether the binding is dependent on ARID1A will be critical to improve our current work and help this field of research move forward.

As such we conducted ChIP of AURKA promoter using antibodies against SNF5 and BRG1, and here provide an evidence that the two core components are indeed recruited to AURKA promoter in CRC cells in an ARID1A-dependent manner. Details of the additional data are described in the Comment #3 (see below). We also cited the two reports in our revised manuscript and discussed their findings together with our observations in the discussion section (described in the Comment #3).

We also believe that our key finding - CDC25C axis is a convergent point where ARID1A mutation and AURKA activation signals meet and is constitutively active in CRC cells with ARID1A mutation, making the cells addicted to the oncogenic signaling, and thereby becoming a synthetic lethality target in ARID1A-deficient CRC - further strengthen the novelty of our study.

We are again grateful to the reviewer to point out that we did not compare efficacies of other known synthetic lethality targets with AURKA. Based on the suggestions, we carefully re-analyzed the screening raw data to check why the known inhibitors were absent from the results and further tested inhibitors of four known synthetic lethality targets in ARID1A isogenic CRC pair. Details of the additional data are described below in the Comment #1.

Lastly, we added current development status of AURKA inhibitors in clinical trials and discussed future perspectives of AURKA inhibitors in cancer treatment in the discussion section, shown as following:

“...A number of recent studies have shown that AURKA is frequently overexpressed in several tumors, including leukemia²⁸, colorectal³⁴, ovarian²⁹ and pancreatic tumors³². Several small molecule kinase inhibitors targeting AURKA such as alisertib, danusertib, MK-5108, and ENMD-2076, have entered clinical trials for cancer treatment (<https://clinicaltrials.gov/>). Alisertib (MLN8237) is the most advanced AURKA inhibitor in clinical setting and is currently under Phase I/II/III clinical investigations for leukemia and many other solid tumors⁵⁹. Clinical efficacies of alisertib vary depending on tumor types and some cases of serious side effects have been described⁶⁰. However, potential clinical effect of alisertib is promising as it improved progression-free survival and the duration of disease stability in various tumor types, and the reported side effects were manageable in many cases⁶¹. To date, clinical studies of AURKA inhibitors in hematologic malignancies have moved fast, but there has been a slow progress in solid tumors. Therefore, prompt clinical investigations of AURKA inhibitors for colorectal and ovarian cancers with ARID1A deficiency where AURKA is highly expressed are warranted...”

Although the main conclusion of this study is AURKA-CDC25C axis as a cancer target in CRC with ARID1A loss, our mechanistic study provides a novel genetic/functional interaction between

ARID1A (or SWI/SNF nucleosome remodeler) and AURKA in induction of an oncogene addiction in cells with ARID1A mutation. Therefore we believe that the study has broad interest among communities of basic cancer research, drug discovery and clinical & translational research, and would be reasonable for consideration for publication in Nat Comm. This is evidenced by several recent papers reporting identifications of SWI/SNF related synthetic lethality targets that have been published in broad-range scientific/biomedical journals, such as PNAS (Hoffman et al, 2014 for BRG1 and BRM synthetic lethality), Nat Med (Bitler et al, 2015 for ARID1A and EZH2 synthetic lethality), Nat Comm (Williamson et al, 2016 for ARID1A and ATR synthetic lethality), Nat Comm (Tagal et al, 2017 for BRG1 and AURKA synthetic lethality) and eLIFE (Kelso et al, 2017 for ARID1A and ARID1B synthetic lethality).

We again thank the reviewer for taking time to thoroughly review our work and pointing out several key issues that needed to be addressed. Described below are point-to-point responses to the reviewer's specific comments.

Major Comments:

1) Did the authors observe any of the other previously-identified epigenetic modifiers that act as synthetic lethals for ARID1A loss in their screen in Figure 1D? If so, how did they compare to the AURKA inhibitors? If not, how do they explain their absence?

Response: In the initial screening we mainly focused on the new synthetic lethality candidates that showed the best efficacy in the screening, therefore we did not analyze marginally effective hits, including those known synthetic lethality inhibitors. However, as we agree with the reviewer's comments, we carefully re-analyzed the screening data to locate other known inhibitors in the IC50 plot (Fig. 1e in the revised manuscript), and further tested the efficacy of the known synthetic lethality inhibitors to compare with AURKAi (Fig. 1g in the revised manuscript).

Fig. 1e. A log₁₀-IC₅₀ plot of the screening results. A log₁₀ scale of IC₅₀ values of the drugs against HCT116 ARID1A wildtype and KO cells was plotted. Drugs that were more selective for ARID1A-KO cells were selected as synthetic lethality drugs.

In the screening, PARP inhibitors and HDAC inhibitors showed some sort of selectivity toward ARID1A-KO cells, although they were not as selective as AURKA inhibitors. In the raw data, we found that, many PARP inhibitors used in the screening did not inhibit more than 50% of cell

growth in both cell lines. Similar results were observed in the previous report (Willamson et al, ATR inhibitors as a synthetic lethality therapy for tumours deficient in ARID1A. 2016, Nat Comm) that olaparib did not inhibit the growth of ARID1A isogenic cells. We therefore tested PARP inhibitor olaparib at higher concentrations (up to 300 μM) with longer incubation time (5 days) and found that it indeed showed the synthetic lethality in ARID1A-isogenic CRC cells (Fig. 1g in the revised manuscript, 4th panel). HDAC6 inhibitor showed a marginal selectivity toward ARID1A-KO cells in the screening and was further confirmed with additional experiments with tubastatin A (Fig. 1g in the revised manuscript, 2nd panel). ATR inhibitor was not included in the epigenetics library, but was a hit from our kinase inhibitor library screening (data not included in this paper). To compare the efficacy of ATR inhibitor with AURKAI we analyzed a dose-response curve for an ATR inhibitor VE821 in the ARID1A-isogenic CRC cells. VE821 showed a comparable synthetic lethality effect with AURKAI in the ARID1A-isogenic CRC cells (Fig. 1g in the revised manuscript, 1st and 3rd panels). Finally, EZH2 inhibitors were not shown as hits in our screening and this result was further verified with a dose-response curve of EPZ-6438, a selective EZH2 inhibitor. Up to 300 μM treatment, EPZ-6438 did not show synthetic lethality effects in the ARID1A isogenic HCT116 cells (Fig. 1g in the revised manuscript, 5th panel). To explore a possible reason for this result, we analyzed the expression status of PI3K-interacting protein 1 (PIK3IP1) in the ARID1A isogenic CRC cells. In ovarian cancer cells, PIK3IP1 is a direct target gene of the ARID1A-EZH2 synthetic lethality in which EZH2 represses PIK3IP1 expression in cells with ARID1A loss and inhibition of EZH2 activity restores PIK3IP1 to induce apoptosis in the cells (Bitler et al, 2015, Nat Med). In CRC cells, however, ARID1A loss did not reduce PIK3IP1 expression (Supplementary Fig. 2d), suggesting a possible reason for not showing synthetic lethality effect by EZH2 inhibitor in ARID1A-KO CRC cells.

Fig. 1g. Dose response curves of HCT116 ARID1A isogenic cell pair treated with AURKAI and known synthetic lethality compounds for ARID1A are shown.

Supplementary Fig. 2d, Western blot analysis of ARID1A and PIK3IP1 expression in ARID1A-isogenic HCT116 cell lines.

Based on the results, we added following description in the result section of the revised manuscript: “...From two rounds of screening, we identified three aurora kinase A (AURKA) inhibitors, including Aurora A Inhibitor I (AURKAi), MK-5108 and JNJ-7706621, as selective inhibitors of ARID1A-KO cells over the wildtype ones (Fig. 1e; Supplementary Fig. 2c). Inhibitors of PARP, HDAC and histone demethyltransferase (HDMT) were also shown up as selective drugs, although they were not as selective as AURKA inhibitors in this screening. Since AURKAi showed the highest selectivity against ARID1A KO cells among the identified hits, we selected AURKAi as a main synthetic lethality compound for follow-up studies. AURKAi treatment showed decent selectivity toward all three ARID1A-KO clones compared to wildtype HCT116 cells (Fig. 1f). We also compared the synthetic lethality effect of AURKAi with other known ARID1A synthetic lethality targets, including HDAC6²¹, ATR²⁰, PARP¹⁹, and EZH2¹⁸. The synthetic lethality effect of AURKAi was largely comparable with inhibitors of HDAC6, ATR, and PARP (Fig. 1g). EZH2 inhibitor, EPZ-6438 did not show synthetic lethality effect in the HCT116 CRC isogenic pair. This was possibly because ARID1A deficiency in colorectal cancer cells did not reduce PI3K-interacting protein 1 (PIK3IP1) expression, a key target gene for the synthetic lethality by EZH2 inhibitors in ovarian cancer cells (Supplementary Fig. 2d)...”

2) The authors propose that ARID1A acts as a major regulator of AURKA expression. They should show the levels of AURKA protein for the cell lines in Figure 3a as a further test of that model.

Response: Thanks to the reviewer’s comment, we noticed that this is an important point we missed from the initial study. Therefore, we have analyzed AURKA expression in the 6 CRC cell panel along with the ARID1A expression level. The result clearly indicated an inverse correlation of expression status between ARID1A and AURKA in the CRC cell panel, with ARID1A-deficient RKO cells showing the highest level of AURKA (Fig. 5m in the revised manuscript). For smooth explanation of the data, we incorporated this data into Fig. 5 in the revised manuscript where we show the transcription repression of AURKA by ARID1A.

Fig. 5m. Protein expression status of ARID1A and AURKA in six colorectal cancer cell panel.

3) They should assess the binding of at least 2 other core members of the SWI/SNF complex by ChIP as shown for ARID1A in Figure 5g. In other words, is the SWI/SNF complex present at the TSS of AURKA in the absence of ARID1A or does it recruit the complex?

Response: We are again grateful to the reviewer for pointing out this critical issue to broaden our view on the synthetic lethality interaction. Among the core components of the SWI/SNF complex the reviewer suggested, we selected BRG1 and SNF5 for ChIP analysis, because the 2 core components have been reported to have a synthetic lethality interaction with AURKA. We used ChIP grade antibodies for BRG1 and SNF5 and conducted ChIP of AURKA promoter (TSS) in both ARID1A wildtype and KO CRC cells. Similar to that seen in ARID1A ChIP, BRG1 and SNF5 indeed bind to the promoter of AURKA, and the binding was not observed in ARID1A-KO cells (Fig. 5h and i in the revised manuscript).

Fig. 5h and i. ChIP of AURKA promoter in HCT116 ARID1A wild type (black) and KO (gray) using anti-BRG1 antibody. ** $P < 0.01$ vs IgG, One sample t -test. (i) ChIP of AURKA promoter in HCT116 ARID1A wild type (black) and KO (gray) using anti-SNF5 antibody. ** $P < 0.01$ vs IgG, One sample t -test.

The result indicates that (1) the core components of SWI/SNF complex are present at the TSS of AURKA and (2) the recruitment of the complex is dependent on ARID1A. Based on this notion, we have added following description in the Results and Discussion sections of the revised manuscript:

“...ARID1A is known to play a key role in targeting of SWI/SNF complex to DNA via its ARID-DNA binding domain⁴². We thus wonder whether other core components of SWI/SNF complex are recruited to AURKA promoter for transcription regulation and, if so, whether the recruitment is dependent on ARID1A. Hence, we next analyzed ChIP assay of AURKA promoter using antibodies against two core components of SWI/SNF, BRG1 (SMARCA4) and SNF5 (SMARCB1) in ARID1A wildtype and KO HCT116 cells. As shown in the ChIP results, the two core components were indeed recruited to the AURKA promoter in ARID1A wildtype, but not in ARID1A KO, cells (Fig. 5h, i). These data suggest that SWI/SNF complex is recruited to AURKA promoter via ARID1A-dependent targeting and represses the transcription of AURKA in CRC cells...”

“...Mechanistically in CRC cells ARID1A loss enhanced AURKA transcription, making the cells addicted to AURKA signaling for their growth and survival. This observation is further strengthened by recent two reports showing that two other components of SWI/SNF complex are involved in the down-regulation of AURKA expression and cancer cell sensitivity to AURKA inhibitors. Lee et al.⁵⁴, reported that SNF5, a component of SWI/SNF represses AURKA transcription in rhabdoid tumors. AURKA was overexpressed in SNF5-mutant rhabdoid tumors and AURKA silencing sensitized the tumor cells to induce apoptosis⁵⁴. More recently, Tagal et al.⁵⁵, showed that AURKA is essential for the survival of non-small cell lung cancer cells that harbor inactivation mutations of BRG1, another SWI/SNF component protein. However, it was unclear whether each component of SWI/SNF complex participates in the synthetic lethality independently or they work as a complex. Our ChIP analysis of AURKA promoter with antibodies against the two core components in ARID1A WT and KO cells demonstrated that the targeting of SNF5 and BRG1 to the AURKA promoter is dependent on ARID1A. ARID1A contains a DNA-binding (ARID) domain and is known to play a key role in targeting the complex to the target gene promoter⁴². On the other hand, SNF5 is essential for the formation of SWI/SNF complex⁵⁶ and BRG1 provides energy derived from ATP hydrolysis to the complex for the nucleosome remodeling activity⁵⁷. Given the essentiality of the three components in the nucleosome remodeling and transcription regulation functions of SWI/SNF complex, it becomes apparent that entire SWI/SNF complex has a synthetic lethality interaction with AURKA in tumor cells where mutations in key components of SWI/SNF complex cause an induced essentiality or oncogene addiction of AURKA for the cell survival...”

In addition to the data addressing the three major comments, we have included several additional data in the revised manuscript to improve the quality of the study, such as generation of additional ARID1A-isogenic CRC (SW480) pairs and testing synthetic lethality effects, synthetic lethality in heterozygous ARID1A-KO with loss of protein expression, synthetic lethality in ovarian cancer cell lines, and inclusion of more controls in ChIP experiments. We believe that the revised manuscript has been substantially improved by addressing most, if not all, of the reviewer's concerns.

Reviewers' comments:

Reviewer #1 (Remarks to the Author):

The revision is thorough and has answered all of our questions fully. Notably, they have generated multiple het and homo cell lines and have validated the synthetic lethal relationship with Aurki.

Reviewer #3 (Remarks to the Author):

The authors describe an impressive suite of drug and functional studies that describe a synthetic lethal relationship between ARID1A and AURKA in CRC models, identify the phenotypic results of this dependency on the G2/M checkpoint, elucidate convergence of ARID1A and AURKA signaling on CDC25C, and provide evidence to support selective anticancer activity of AURKA inhibitors in the setting of ARID1A-mutant CRC and possibly ovarian carcinoma as well.

The studies are well-designed and informative. Key concerns of Reviewer #1 have been clearly addressed. Reviewer #2's overarching concern in regards to broad impact and novelty has been partially addressed through extended ChIP experiments, and some additional description of models and drug studies. Some concerns over broader impact do remain given that the authors have chosen to perform the majority of their work in CRC models with a focused addition of OC studies in a few limited cell lines. However, these concerns should be addressable through revision of the text without additional experimentation. The value of these findings in the CRC models is indeed of significant interest to a broader audience, but the broader implications of these studies can be better emphasized through addition of text in the Introduction and Discussion that is similar to that the authors provide in the response to Reviewer #2. Some comments below also point to areas that can help clarify novelty and relevance through revisions to the text. Additionally, some details of the drug studies are still needed in order to quantitatively convey the importance of the AURKA hits relative to other known genes and inhibitors of interest in ARID1A-mutant cancers (also referenced in comments below). Finally, some systematic concerns over writing style and quality include a general imbalance of appropriate detail in Introduction, Results, and Methods in addition to grammatical errors and some lack of clarity. These points may be addressed with editorial assistance at the editor's discretion, but are partially detailed in the minor points below.

Major points:

1. The Results and Figure 1 legend still don't adequately describe the drug studies and prioritization of AURKA inhibitors. Unanswered questions include: How many ARID1A KO clones were evaluated in the screen and how did the results compare between clones? How many drugs were more selective for the ARID1A KO than WT? Quantitatively, how did the AURKA inhibitors match up against the other ARID1A KO selective drugs?
2. A major concern is that all drug study endpoints appear to have been assessed at 72 hours. Epigenetic agents often show longer-term effects that are important to consider (e.g. five-day endpoints sometimes yield different results). These timepoint considerations may underlie puzzling results such as the lack of differential activity of the EZH2 inhibitor and would be important to help contextualize the potency of AURKA inhibition against other known targets/modalities. This concern could be mostly addressed through targeted assessment of five-day effects of the shortlist of selective agents.
3. Did the authors generate both homozygous and heterozygous KOs of HCT116? Similarly to experiments with SW480, did they assess differential effects of AURKA inhibition or knockdown in hets versus homozygotes? Why did the authors choose to do these experiments in SW480 rather than HCT116? Experiments in all of these setting are not absolutely necessary, but better description and defense of approach is needed.
4. The description of the ChIP experiments needs additional detail/clarification and the below points need to be addressed: Why were different primer pairs chosen as controls for ChIP rather than primers at other loci? Why did the authors normalize to IgG rather than to a gene not

regulated by the gene of interest or to a non-transcribed locus? Was quantification calculated as % input (input was not discussed at all) or $\Delta\Delta Ct$? These elements should be clearly represented in the Results, Methods, and Figures. Additionally, some ChIP figures appear to be normalized/compared to IgG and others are normalized/compared to a control cell line. Finally, a word of caution - mouse antibodies are known to have a high background of SWI/SNF binding to IgG such that signal may have been masked.

5. The authors should more clearly defend the selection of ovarian cancer cell lines and the significance of these data both in the Introduction and Results since these data are a crucial link to potentially broader relevance of their findings in CRC models. ARID1A mutations are quite common in ovarian clear cell carcinoma (OCCC), but none of the studied cell lines are OCCC. Ovarian serous carcinomas tend to bear a high mutation burden amidst p53 loss and genomic instability and thus may be confounding.

6. Although arguably beyond the scope of this study, additional genomic or epigenomic profiling of the HCT116 isogenic model would be of significant value. Certainly at least RNA sequencing if not also ATAC-seq are readily available techniques that would add tremendous value to prioritization of molecular dependencies in the setting of ARID1A loss.

Minor points:

1. Independently of the quality of the science, a significant need exists for improvements to both grammar and clarity. Key details are also occasionally missing or oddly distributed between Results, Methods, and Figures. See examples of these needs below.

a. Diction/grammar needs to be closely checked for completeness or clarity in a number of areas. For example:

i. Lines 16 and 35 need a determiner in front of "SWI/SNF chromatin remodeling complex"

ii. Line 18 "inactivation" should be "inactivating"

iii. Line 19 "has been necessitated" isn't quite right

iv. Line 23 "targeted to" should be "occupied"

v. Line 36 - the SWI/SNF complex doesn't "contain" two major subclasses, but instead can exist as two major forms

vi. Line 39 - mutations in SWI/SNF subunits don't "induce various genetic alterations"

vii. Many more...

b. A number of areas also exist in which a need exists for some brief, but critical contextual detail, particularly in the Introduction and Discussion:

i. Lines 48-50 are missing information on how ARID1A deficiency was found to impact TSG functions in these references.

ii. Lines 51 and 52 - Describing ARID1A deficiency as a "target" is not quite right even though the general concept can be followed. Similarly, what do the authors mean by "synthetic lethality"? It is described as "an approach to ... targeting of cancer cells harboring the tumor suppressor mutation" when in fact synthetic lethality is an interdependency between genes that may be exploited in certain therapeutic strategies. While this may seem to be a minor point, it is important for contextualizing understanding of cell survival dependencies and selective therapeutic targeting of these dependencies in the setting of ARID1A loss. It seems clear that the authors understand these nuances, but it is important to convey them with precision.

iii. Lines 58 and 59 - ARID1A is itself a component of epigenetic machinery, so why are ARID1A's epigenetic synthetic lethal relationships so surprising?

iv. Lines 60 - 63 - A brief description of why/how AURKA was prioritized apart from "novelty" is critical here.

v. Many more...

2. Minor additional details are also needed in several areas:

a. The authors clearly describe the high frequency and breadth of ARID1A mutations across cancers, but specific description of overall frequency across cancers and/or frequency in CRC and OC would be helpful.

b. Why did the authors focus on CRC and OC? An additional defense of why findings in CRC may be relevant in other cancers would be helpful.

c. What additional driver mutations (genomic background) are present in HCT116 that might play

a role in results seen with the screens? In particular, are other SWI/SNF or epigenetic mutations present? It's similarly important in interpretation of results to consider these issues for other models utilized.

d. It's important to clarify which models are ARID1A het mutants versus homo mutants. Incorporation of standard terminology is needed – e.g. rather than "KO HCT116 ARID1A" use "HCT116-ARID1A(x/y)" where "x" and "y" are the specific AA mutation or "+" if WT.

e. What dose was used for xenograft longitudinal measurement studies? Presumably 60 mg/kg given that this is the only dose that reduces tumor weight. However, it's also not clear at what timepoint tumor weight was measured.

f. How do the IC50s and doses tested in mice hold up against known human PK and tox data for the AURKA inhibitors? Are the effective doses in ranges that suggest we might be able to achieve sufficient concentrations in tumors to see an effect?

Author responses to Referees

Reviewers' comments:

Reviewer #1 (Remarks to the Author):

The revision is thorough and has answered all of our questions fully. Notably, they have generated multiple het and homo cell lines and have validated the synthetic lethal relationship with Aurki.

Response: We appreciate the reviewer for the time and consideration for our manuscript. We are glad that the reviewer is satisfied with the revision.

Reviewer #3 (Remarks to the Author):

The authors describe an impressive suite of drug and functional studies that describe a synthetic lethal relationship between ARID1A and AURKA in CRC models, identify the phenotypic results of this dependency on the G2/M checkpoint, elucidate convergence of ARID1A and AURKA signaling on CDC25C, and provide evidence to support selective anticancer activity of AURKA inhibitors in the setting of ARID1A-mutant CRC and possibly ovarian carcinoma as well.

The studies are well-designed and informative. Key concerns of Reviewer #1 have been clearly addressed. Reviewer #2's overarching concern in regards to broad impact and novelty has been partially addressed through extended ChIP experiments, and some additional description of models and drug studies. Some concerns over broader impact do remain given that the authors have chosen to perform the majority of their work in CRC models with a focused addition of OC studies in a few limited cell lines. However, these concerns should be addressable through revision of the text without additional experimentation. The value of these findings in the CRC models is indeed of significant interest to a broader audience, but the broader implications of these studies can be better emphasized through addition of text in the Introduction and Discussion that is similar to that the authors provide in the response to Reviewer #2. Some comments below also point to areas that can help clarify novelty and relevance through revisions to the text. Additionally, some details of the drug studies are still needed in order to quantitatively convey the importance of the AURKA hits relative to other known genes and inhibitors of interest in ARID1A-mutant cancers (also referenced in comments below). Finally, some systematic concerns over writing style and quality include a general imbalance of appropriate detail in Introduction, Results, and Methods in addition to grammatical errors and some lack of clarity. These points may be addressed with editorial assistance at the editor's discretion, but are partially detailed in the minor points below.

Response: We have checked entire manuscript based on the Reviewer's comments and found that several parts of the writing are either unclear or miswritten. We sincerely appreciate the Reviewer for constructive comments and detailed suggestions to improve the quality of our manuscript. We carefully addressed all the Reviewer's concerns and revised the manuscript accordingly. Described

below are point-to-point responses to the Reviewer's comments. Descriptions in black color are Reviewer's comments, Red color are our answers and those in blue color are revised texts in the manuscript.

Major points:

1. The Results and Figure 1 legend still don't adequately describe the drug studies and prioritization of AURKA inhibitors. Unanswered questions include: How many ARID1A KO clones were evaluated in the screen and how did the results compare between clones? How many drugs were more selective for the ARID1A KO than WT? Quantitatively, how did the AURKA inhibitors match up against the other ARID1A KO selective drugs?

Response: We used KO clone #1 for the screen (Fig. 1d and e) and used other clones (# 1-3) for the validation of screening hit (Fig. 1f). As the reviewer suggested, we clarified this in the Results (Results, page 5) "...Among the three confirmed ARID1A KO clones (ARID1A^{-/-} #1-3) (Fig. 1c), ARID1A^{-/-} #1 was used for the synthetic lethality screening and the other clones were used to validate the screening hits.") and Figure 1 & legend in the revised manuscript.

For the prioritization of the screening hits, we used selectivity index (SI): $IC_{50}^{ARID1A(+/+)} / IC_{50}^{ARID1A(-/-)}$ to prioritize synthetic lethality candidates. Among the epigenetics compounds screened, 6 drugs have SI>2. AURKA inhibitors were the majority (3 out of 6) among the identified candidates. We have described this prioritization in the Results (page 5 (Results, page 5) "...From two rounds of screening, we identified 6 candidate drugs that showed a selectivity index (SI) larger than 2 for the ARID1A^{-/-} #1 cells; the candidates included 3 AURKA inhibitors (Aurora A Inhibitor I, MK-5108 and JNJ-7706621), a histone demethyltransferase (HDMT) inhibitor (GSK J4), a PARP inhibitor (PJ34), and a histone methyltransferase (HMT) inhibitor (BIX 01294) (Fig. 1e; Supplementary Fig. 2c and d). Since the majority of the identified candidates were AURKA inhibitors (3 out of 6), we selected Aurora A Inhibitor I (AURKAi) as the primary synthetic lethality compound for follow-up studies..."), Figure 1 legend, Methods and Supplementary Data in the revised manuscript.

Supplementary Figure 2 (c) Selectivity index (SI) of the synthetic lethality candidates for ARID1A. $SI = IC_{50}^{ARID1A(+/+)} / IC_{50}^{ARID1A(-/-)}$. Among all the epigenetics compounds tested, top 10 candidates are shown in the graph and drugs with SI>2 are indicated.

Figure 1 (e) A log₁₀-IC₅₀ plot of the screening results. A log₁₀ scale of IC₅₀ values of the drugs against HCT116 ARID1A wildtype and KO cells was plotted. Drugs with selectivity index (SI) > 2 for ARID1A^{-/-} cells were selected and marked as synthetic lethality candidates.

2. A major concern is that all drug study endpoints appear to have been assessed at 72 hours. Epigenetic agents often show longer-term effects that are important to consider (e.g. five-day endpoints sometimes yield different results). These timepoint considerations may underlie puzzling results such as the lack of differential activity of the EZH2 inhibitor and would be important to help contextualize the potency of AURKA inhibition against other known targets/modalities. This concern could be mostly addressed through targeted assessment of five-day effects of the shortlist of selective agents.

Response: We agree with the reviewer’s suggestion that epigenetics drugs may show long-term effects in the cells. Therefore we conducted 5-day cell viability experiments with the five known synthetic lethality compounds. All the drugs were overall more potent in 96 h treatment compared to 72 h, but IC₅₀ for BOTH cell lines shifted to the left. Thus, the synthetic lethality effects of the five compounds between the two treatment time points were overall similar. The EZH2 inhibitor IC₅₀ values were also shifted to the left in 96 h treatment. But, similar to other drugs, this happened in both WT and KO cell lines, thus no synthetic lethality effect by EZH2 inhibitor was observed in a long-term treatment. Therefore, we believe that the lack of the synthetic lethality effect of the EZH2 inhibitor in CRC cells is likely because the PIK3IP1 expression is not directly regulated by ARID1A in CRC cells (we provided the PIK3IP1 expression level in ARID1A WT and KO CRC cells in the Supplementary data and described this observation in the manuscript, page 5). Another possible reason could be the PIK3CA activating mutation in HCT116 cells. We described the 96 h treatment outcome of EZH2 inhibitor in ARID1A isogenic CRC cells in the revised manuscript (Results, page 5) “The EZH2 inhibitor EPZ-6438 did not have synthetic lethality effects on the HCT116 CRC isogenic pair (Fig. 1k), even with longer (96 h) treatment (data not shown). This effect was possibly because ARID1A deficiency in CRC cells did not reduce the expression of PI3K-interacting protein 1 (PIK3IP1), which is a key target gene for the synthetic lethality of EZH2 inhibitors in ovarian cancer cells (Supplementary Fig. 2e); alternatively, the effect could be due to the activating mutation in PIK3CA in HCT116 cells³⁵)” to clarify the lack of differential sensitivity is not due to the short treatment time. The PIK3CA mutation in HCT116 is further discussed in the Discussion section (page 12, will be discussed again at the Reviewer’s minor comment #2c below).

3. Did the authors generate both homozygous and heterozygous KOs of HCT116? Similarly to experiments with SW480, did they assess differential effects of AURKA inhibition or knockdown

in hets versus homozygotes? Why did the authors choose to do these experiments in SW480 rather than HCT116? Experiments in all of these setting are not absolutely necessary, but better description and defense of approach is needed.

Response: Since we failed to isolate ARID1A^{+/-} clones from HCT116 ARID1A KO study, we conducted additional KO study in another CRC cell line, SW480 where we successfully obtained both ARID1A^{-/-} and ARID1A^{+/-} clones. As the reviewer pointed out, we clarified this in the Results section of the revised manuscript, as shown below:

(Results, page 6-7) “Because many of the ARID1A mutations in patients are heterozygous (ARID1A^{+/-})^{3,38}, it is important to test whether the synthetic lethality of AURKAi is also effective in cells with heterozygous mutations. Since we failed to isolate ARID1A^{+/-} clones from the HCT116 ARID1A KO study, we knocked out ARID1A in SW480 cells, an ARID1A^{+/+} CRC cell line. Through CRISPR/Cas-9 gene editing, we successfully generated both homozygous (ARID1A^{-/-}) and heterozygous (ARID1A^{+/-}) KO clones from SW480 cells (Supplementary Fig. 4a, b).”

4. The description of the ChIP experiments needs additional detail/clarification and the below points need to be addressed: Why were different primer pairs chosen as controls for ChIP rather than primers at other loci? Why did the authors normalize to IgG rather than to a gene not regulated by the gene of interest or to a non-transcribed locus? Was quantification calculated as % input (input was not discussed at all) or $\Delta\Delta Ct$? These elements should be clearly represented in the Results, Methods, and Figures. Additionally, some ChIP figures appear to be normalized/compared to IgG and others are normalized/compared to a control cell line. Finally, a word of caution - mouse antibodies are known to have a high background of SWI/SNF binding to IgG such that signal may have been masked.

Response: We are grateful to the reviewer to point out the lack of clarity in the ChIP experiments and all the helpful suggestions to improve the data presented. Described below are point-to-point responses and relevant revisions:

(1) Why were different primer pairs chosen as controls for ChIP rather than primers at other loci?
Answer: The two control primer pairs we used were designed for far upstream (-3,800bp) or downstream (+3,200bp, intronic region) of the transcription start site (TSS) of AURKA gene, which were suggested by the Reviewer #1. Both primers were not enriched by the ARID1A ChIP (the level similar to IgG control). As the Reviewer suggested, to avoid any unnecessary argument, we re-analyzed ChIP-qPCR using a control primer pair at other loci (ORF-free region) (Tan SK et al, 2011, EMBO J, 30:2569–2581). The control primer locus was not enriched by ARID1A ChIP, while AURKA promoter was significantly enriched (Fig. 5f and 5k in the revised manuscript, also shown below). We replaced the ChIP data with the new one and revised related Text, Figure, and Supplementary data accordingly.

Fig. 5. (f) Chromatin immunoprecipitation (ChIP) of AURKA promoter in ARID1A^{+/+} HCT116 cells using anti-ARID1A antibody. ChIP data were normalized to the control IgG ChIP with the control (CTRL) primer. ** $P < 0.01$, Student's t -test. (k) ChIP of AURKA promoter in ARID1A^{OE} RKO clone #2 using anti-ARID1A antibody. ChIP data were normalized to the control IgG ChIP with the control (CTRL) primer. * $P < 0.05$, Student's t -test.

(2) Why did the authors normalize to IgG rather than to a gene not regulated by the gene of interest or to a non-transcribed locus? Was quantification calculated as % input (input was not discussed at all) or $\Delta\Delta Ct$? **Answer: We used $\Delta\Delta Ct$ method where IgG control ChIP was used to normalize antibody ChIP, according to the manufacturer's instruction. Since the enrichment of the non-transcribed locus (ORF-free region) by ARID1A ChIP was a background level similar to IgG control (shown in Fig. 5f above), we believe that the normalization with IgG control is reasonable. Input was used to calculate ΔCt (normalized ChIP). We thanks to the Reviewer to point out this issue. We clarified the control primer information and ChIP quantification method in the Method section in the revised manuscript, as following:**

(Methods, page 16) "Following are sequences of the primer pairs used to analyze ChIP DNA. AURKA primer: forward primer 5'-ACAGGTCTGGCTGGCCGTTGGC-3' and reverse primer 5'- GGCGCACACCGCGCGCAGGCG-3' (Integrated DNA Technologies). Control primer (ORF-free region): forward primer 5'-CCTGGAGGGCTTGAGATG-3' and reverse primer 5'-GATCCTACGGCTGGCTGTGA-3' (a kind gift of Prof. Edwin Cheung at the University of Macau)⁷⁵. The ChIP-qPCR data were expressed as fold enrichment ($\Delta\Delta Ct$ method) according to the manufacturer's instruction. Briefly, each ChIP DNA fraction's Ct value was normalized to the input DNA fraction Ct value (ΔCt [normalized ChIP]). The ΔCt for antibody ChIP was normalized to IgG control ChIP ΔCt ($\Delta\Delta Ct$ [antibody ChIP/IgG ChIP]). Fold enrichment of the specific site was calculated according to the following equation: fold enrichment = $2^{-\Delta\Delta Ct}$ [antibody ChIP/IgG ChIP]."

(3) Some ChIP figures appear to be normalized/compared to IgG and others are normalized/compared to a control cell line. **Answer: As we agree to the Reviewer's comment about**

inconsistent format of the data presentations in Fig 5g and 5l (RNA-Pol II ChIP comparison between 2 cell lines), we normalized each RNA-Pol II ChIP to IgG control (Fig. 5g and 5l shown below, in the revised manuscript).

(g)

(l)

Figure 5. (g) ChIP of AURKA promoter in HCT116 ARID1A^{+/+} and ARID1A^{-/-} cells using anti-RNA-Pol II antibody. IgG in each cell line was used as a normalization control. * $P < 0.05$, Student's t -test. (l) ChIP of AURKA promoter in RKO ARID1A^{-/-} and ARID1A^{OE} cells using anti-RNA-Pol II antibody. IgG in each cell line was used as a normalization control. ** $P < 0.01$, Student's t -test.

(4) Finally, a word of caution - mouse antibodies are known to have a high background of SWI/SNF binding to IgG such that signal may have been masked. **Answer: We thanks Reviewer for the caution. We used the same ChIP-grade mouse antibodies that Bitler et al (Nat Med, 2015; Nat Cell Biol, 2017) used for the ARID1A and BRG1 ChIP. The antibody information is available in the Methods section (page 15). Based on the results with ChIP controls (IgG control and control primer at non-transcribed locus), we observed that the SWI/SNF ChIP of AURKA promoter by the antibodies we used was significantly higher than IgG.**

5. The authors should more clearly defend the selection of ovarian cancer cell lines and the significance of these data both in the Introduction and Results since these data are a crucial link to potentially broader relevance of their findings in CRC models. ARID1A mutations are quite common in ovarian clear cell carcinoma (OCCC), but none of the studied cell lines are OCCC. Ovarian serous carcinomas tend to bear a high mutation burden amidst p53 loss and genomic instability and thus may be confounding.

Response: We appreciate the reviewer to point out the important aspect of our findings. Firstly we have introduced mutation frequency of ARID1A in ovarian cancer and CRC, and highlighted a clinical significance of ARID1A loss in CRC pathogenesis in the Introduction section:

(Introduction, page 3) “Genome-wide sequencing analyses of tumor samples revealed that 46-57% of OCCC cases harbored loss-of-function mutations in the ARID1A gene, implying the significant contribution of aberrant ARID1A functions to OCCC pathogenesis^{3, 12}. In CRC patients, a mutation frequency of approximately 10% was observed for the ARID1A gene¹³. However, clinic-pathological analyses of ARID1A protein levels in CRC tumor samples showed that 25.8% of CRC primary tumors did not express ARID1A, and 51.2% had low expression levels of ARID1A (77% of all the CRC samples had no or low ARID1A expression)¹⁴. The loss of ARID1A expression became even more significant as the tumor-node-metastasis (TNM) stage advanced. ARID1A loss was observed for 7.4% of TNM stage I samples, 24.1% of TNM stage II samples, 22.2% of TNM stage III samples, and 46.3% of TNM stage IV samples¹⁴. These data suggest that ARID1A loss in CRC is strongly associated with tumor progression and metastasis.”

Second, we described the selection of ovarian cancer cells and the significance of the data in the Results section:

(Results, page 7) “Since ARID1A mutations are highly common in OCCC, we next tested synthetic lethality in different subtypes of ovarian carcinoma cell lines. ARID1A mutant SKOV3 cells were originally described as high-grade serous carcinoma, but it was recently re-described as OCCC according to histological and immunological characterizations of in vivo tumors^{39, 40}. ES-2 cells express wild-type ARID1A and were originally described as OCCC^{41, 42}. HO8910 cells characterized as ovarian serous carcinoma with wild-type ARID1A. Our results showed that ARID1A-deficient SKOV3 cells were significantly more sensitive to AURKAi treatment than ARID1A wild-type HO8910 or ES-2 cells (Supplementary Fig. 5a-c). These data suggested that ARID1A-AURKA synthetic lethality exists in ovarian cancer cells and is dependent on ARID1A status rather than on tumor subtype.”

Third, the significance of our findings for broader relevance has been described in the Discussion section with rearrangement of paragraphs in pages 10-11.

(Discussion, pages 10-11) “Mechanistically, ARID1A loss in CRC cells enhanced AURKA..... Our findings are further strengthened by recent two reports showing that two other components of the SWI/SNF complex are involved in the down-regulation of AURKA expression and cancer cell sensitivity to AURKA inhibitors. Lee et al.⁵⁷, reported that SNF5, a component of SWI/SNF, represses AURKA transcription in rhabdoid tumors. AURKA is overexpressed in SNF5 mutant rhabdoid tumors, and AURKA silencing sensitized the tumor cells to apoptosis induction⁵⁷. More recently, Tagal et al.⁵⁸, showed that AURKA is essential for the survival of non-small cell lung cancer (NSCLC) cells that harbor inactivation mutations in BRG1, another SWI/SNF component protein. However, it is unclear whether each component of the SWI/SNF complex causes the synthetic lethality independently or whether they work as a complex. Our ChIP analysis of the AURKA promoter with antibodies against the two core components in ARID1A^{+/+} and ARID1A^{-/-} cells demonstrated that SNF5 and BRG1 targeting to the AURKA promoter is dependent on ARID1A. ARID1A contains a DNA-binding (ARID) domain and is known to play a key role in

targeting the complex to the target gene promoter⁴⁴. On the other hand, SNF5 is essential for the formation of the SWI/SNF complex⁵⁹, and BRG1 provides energy derived from ATP hydrolysis to the complex for the nucleosome remodeling activity⁶⁰. Given the essentiality of the three components in nucleosome remodeling and the transcription regulation functions of the SWI/SNF complex, it is apparent that the entire SWI/SNF complex has a synthetic lethality interaction with AURKA in tumor cells in which mutations in the key components of the SWI/SNF complex causes the induced essentiality or the oncogene addiction of AURKA for cell survival. The observed synthetic lethality in CRC and ovarian cancer models, together with the reported synthetic lethality interactions between other SWI/SNF components and AURKA in rhabdoid tumors and NSCLC models, clearly indicate the potentially broad relevance of our findings to other cancer types where defective SWI/SNF components exist. Indeed, at least 5 components of the SWI/SNF complex, including SNF5, BAF180, ARID1A, BRG1 and BRD7, have been reported to be frequently mutated in a variety of tumor types, such as familial schwannomatosis (30-40% mutation frequency in SNF5), small-cell hepatoblastomas (36% mutation frequency in SNF5), epithelioid sarcomas (55% mutation frequency in SNF5), renal cell carcinoma (41% mutation frequency in BAF180), endometrioid carcinoma (35% mutation frequency in ARID1A), and medulloblastoma (3% mutation frequency in ARID1A and BRG1), in addition to CRC, NSCLC, ovarian and rhabdoid tumors (reviewed by Wilson and Roberts)¹.”

6. Although arguably beyond the scope of this study, additional genomic or epigenomic profiling of the HCT116 isogenic model would be of significant value. Certainly at least RNA sequencing if not also ATAC-seq are readily available techniques that would add tremendous value to prioritization of molecular dependencies in the setting of ARID1A loss.

Response: Thanks for the good suggestion that we are considering in follow up studies. We are planning to screen an RNAi library targeting epigenetics machineries and to analyze transcriptome profiles in the ARID1A isogenic CRC pair and SWI/SNF ChIP-Seq profiling to identify new molecular dependencies in cells with ARID1A loss.

Minor points:

1. Independently of the quality of the science, a significant need exists for improvements to both grammar and clarity. Key details are also occasionally missing or oddly distributed between Results, Methods, and Figures. See examples of these needs below.

a. Diction/grammar needs to be closely checked for completeness or clarity in a number of areas. For example:

i. Lines 16 and 35 need a determiner in front of “SWI/SNF chromatin remodeling complex”

: ‘the’ added

ii. Line 18 “inactivation” should be “inactivating”

: corrected

iii. Line 19 “has been necessitated” isn’t quite right

: the sentence was modified to “ARID1A deficiency has been exploited therapeutically for treating cancer.”

iv. Line 23 “targeted to” should be “occupied”

: corrected

v. Line 36 – the SWI/SNF complex doesn’t “contain” two major subclasses, but instead can exist as two major forms

:corrected

vi. Line 39 – mutations in SWI/SNF subunits don’t “induce various genetic alterations”

:corrected to “Mutations in these subunits lead to the aberrant control of lineage-specific differentiation and gene expression/repression, thereby contributing to tumorigenesis.”

vii. Many more...

b. A number of areas also exist in which a need exists for some brief, but critical contextual detail, particularly in the Introduction and Discussion:

i. Lines 48-50 are missing information on how ARID1A deficiency was found to impact TSG functions in these references.

: for smooth writing, this sentence was removed during the revision.

ii. Lines 51 and 52 – Describing ARID1A deficiency as a “target” is not quite right even though the general concept can be followed. Similarly, what do the authors mean by “synthetic lethality”? It is described as “an approach to ... targeting of cancer cells harboring the tumor suppressor mutation” when in fact synthetic lethality is an interdependency between genes that may be exploited in certain therapeutic strategies. While this may seem to be a minor point, it is important for contextualizing understanding of cell survival dependencies and selective therapeutic targeting of these dependencies in the setting of ARID1A loss. It seems clear that the authors understand these nuances, but it is important to convey them with precision.

: introduced the synthetic lethality concept in more detail, (Introduction, page 3) "...ARID1A deficiency has been exploited therapeutically for treating cancer according to an approach called synthetic lethality. Synthetic lethality is a genetic interaction between two or more genes where a single gene deficiency does not affect cell viability, but the combination of both gene deficiencies causes lethality. This concept has been widely exploited in cancer therapy because many types of cancer have loss-of-function mutations in tumor suppressor genes that are not readily targetable. The pharmacological or genetic disruption of a synthetic lethality target of a tumor suppressor will cause selective lethality in the cancer cells that harbor the tumor suppressor mutations¹⁵."

iii. Lines 58 and 59 – ARID1A is itself a component of epigenetic machinery, so why are ARID1A's epigenetic synthetic lethal relationships so surprising?

: modified to: (Introduction, page 3) "...Recent studies have shown that ARID1A has a synthetic lethality interaction with genes involved in some epigenetic machinery, including EZH2¹⁶, PARP1¹⁷, ATR¹⁸ and HDAC6¹⁹. Inhibiting the synthetic lethality targets resulted in selective vulnerabilities in ARID1A mutant OCCC, CRC and breast cancer cells^{16, 17, 18, 19}. These studies suggested that ARID1A, as an epigenetic machinery component, may have various genetic and functional interdependencies with other epigenetic components to affect cell survival. Based on this notion, we initiated..."

iv. Lines 60 – 63 – A brief description of why/how AURKA was prioritized apart from "novelty" is critical here

: modified to: (Introduction, page 3-4) "...we initiated a systematic screening for druggable targets among human epigenetic machinery using an ARID1A isogenic CRC pair and epigenetics drug library. Among the epigenetics drugs screened, aurora kinase A (AURKA) inhibitors composed the majority of the synthetic lethality hits..."

v. Many more...

2. Minor additional details are also needed in several areas:

a. The authors clearly describe the high frequency and breadth of ARID1A mutations across cancers, but specific description of overall frequency across cancers and/or frequency in CRC and OC would be helpful.

: as we mentioned above (comment #5), we described more details of mutation frequency of ARID1A in CRC and OC in the Introduction section. (page 3) "...Genome-wide sequencing analyses of tumor samples revealed that 46-57% of OCCC cases harbored loss-of-function mutations in the ARID1A gene, implying the significant contribution of aberrant ARID1A functions to OCCC pathogenesis^{3,12}. In CRC patients, a mutation frequency of approximately 10% was observed for the ARID1A gene¹³. However, clinic-pathological analyses of ARID1A protein levels in CRC tumor samples showed that 25.8% of CRC primary tumors did not express ARID1A,

and 51.2% had low expression levels of ARID1A (77% of all the CRC samples had no or low ARID1A expression)¹⁴. The loss of ARID1A expression became even more significant as the tumor-node-metastasis (TNM) stage advanced. ARID1A loss was observed for 7.4% of TNM stage I samples, 24.1% of TNM stage II samples, 22.2% of TNM stage III samples, and 46.3% of TNM stage IV samples¹⁴. These data suggest that ARID1A loss in CRC is strongly associated with tumor progression and metastasis.”

b. Why did the authors focus on CRC and OC? An additional defense of why findings in CRC may be relevant in other cancers would be helpful.

: as described above, high frequency of ARID1A mutations in OC and a stage-dependent, loss of ARID1A expression in a large portion of CRC led us to focus on these 2 cancers. As shown in the response to the Comment #5, we described that our findings in CRC may be relevant to other cancers in the Discussion section.

c. What additional driver mutations (genomic background) are present in HCT116 that might play a role in results seen with the screens? In particular, are other SWI/SNF or epigenetic mutations present? It's similarly important in interpretation of results to consider these issues for other models utilized.

: we are grateful to the Reviewer to point this important issue we missed from the original study. HCT116 cells have PIK3CA hotspot mutation (H1047R) and BRG1 mutation (L1149P). Described below is our interpretation of results after consideration of the two mutations in this model:

(Discussion section, pages 11-12) “...The HCT116 CRC model used in this study has a BRG1 point mutation (L1149P)⁶¹ and a PIK3CA hotspot mutation (H1047R)³⁵. Since EZH2-ARID1A synthetic lethality is mediated by PIK3IP1, which is an endogenous inhibitor of PIK3CA, the PIK3CA activating mutation in HCT116 cells may be another possible contributor to the synthetic lethality of the EZH2 inhibitor, in addition to the lack of PIK3IP1 expression regulation by ARID1A in this model. The BRG1 mutation (L1149P) in HCT116 cells has been well characterized previously^{61, 62}. This mutation does not affect SWI/SNF complex formation, and the BRG1 mutant complexes remain functional in the presence of BRM, another SWI/SNF component that is homologous and partially redundant to BRG1⁶². Based on the findings of our group and others, BRG1 mutations in HCT116 cells do not affect the synthetic lethality interaction of ARID1A and AURKA or other known targets, including PARP1¹⁷ and ATR¹⁸. However, the BRM compensation for BRG1 deficiency may occur in a gene-specific manner as they have different promoter preferences⁶³. Therefore, it cannot be ruled out that BRG1 mutations could potentially affect the synthetic lethality interaction of ARID1A and other targets in the HCT116 model.”

d. It's important to clarify which models are ARID1A het mutants versus homo mutants. Incorporation of standard terminology is needed – e.g. rather than “KO HCT116 ARID1A” use “HCT116-ARID1A(x/y)” where “x” and “y” are the specific AA mutation or “+” if WT.

: modified ARID1A status to ARID1A^{+/+}, ARID1A^{+/-} or ARID1A^{-/-} throughout the manuscript.

e. What dose was used for xenograft longitudinal measurement studies? Presumably 60 mg/kg given that this is the only dose that reduces tumor weight. However, it's also not clear at what timepoint tumor weight was measured.

: Yes it is 60 mg/kg. The tumor weight was measured at 24 days (HCT116) or 20 days (RKO) after drug injection. This information was added in the Figures and Figure legend.

f. How do the IC50s and doses tested in mice hold up against known human PK and tox data for the AURKA inhibitors? Are the effective doses in ranges that suggest we might be able to achieve sufficient concentrations in tumors to see an effect?

: Aurora A Inhibitor I that we used mainly in this study has not been tested in human yet. Alisertib, another AURKA inhibitor is the most widely studied AURKA inhibitor (introduced in the Discussion section, page 12). It has IC50 values ranging from 10 to 500 nM in cancer cells, showed antitumor effect in mice with daily administration of 30-40 mg/kg and has been studied in human with C_{max} = 48 µg/mL. Human plasma level of alisertib is >100 times higher than its IC50 in cells. The IC50 for Aurora A Inhibitor I is about 1 µM in cancer cells and showed antitumor effect in mice with 60 mg/kg. Therefore we assume that the efficacy by Aurora A Inhibitor I could be comparable to that seen by alisertib.

On behalf of all authors, we again thanks to Reviewer for thorough review and thoughtful suggestions. We believe that the revised version of the manuscript now is substantially improved. All the changes made are marked in Red in the revised manuscript. In addition to the parts the reviewer pointed out, the manuscript was edited by a professional English editing service provided by Springer Nature Author Services.

REVIEWERS' COMMENTS:

Reviewer #3 (Remarks to the Author):

The authors have adequately addressed all key concerns. One minor remaining recommendation would be to include the supplemental diagram from the prior version (seems to have been removed in this version) showing the primer sets up- and down-stream of the AURKA TSS for the CHIP data.